# A few-shot Label Unlearning in Vertical Federated Learning

## Abstract

This paper addresses the critical challenge of unlearning in Vertical Federated Learning (VFL), an area that has received limited attention compared to horizontal federated learning. We introduce the first approach specifically designed to tackle label unlearning in VFL, focusing on scenarios where the active party aims to mitigate the risk of label leakage. Our method leverages a limited amount of labeled data, utilizing manifold mixup to augment the forward embedding of insufficient data, followed by gradient ascent on the augmented embeddings to erase label information from the models. This combination of augmentation and gradient ascent enables high unlearning effectiveness while maintaining efficiency, completing the unlearning procedure within seconds. Extensive experiments conducted on diverse datasets, including MNIST, CIFAR10, CIFAR100, and ModelNet, validate the efficacy and scalability of our approach. This work represents a significant advancement in federated learning, addressing the unique challenges of unlearning in VFL while preserving both privacy and computational efficiency.

## 1 Introduction

Vertical Federated Learning (VFL) (Yang et al., 2019) allows multiple organizations to collaboratively utilize their private datasets in a privacy-preserving manner, even when they share some sample IDs but differ significantly in terms of features. In VFL, there are typically two types of parties: (i) the passive party, which holds the *features*, and (ii) the active party, which possesses the *labels*. VFL has seen widespread application, especially in sensitive domains like banking, healthcare, and e-commerce, where organizations benefit from joint modeling without exposing their raw data (Yang et al., 2019; Li et al., 2020).

A fundamental requirement in VFL is the necessity for unlearning, which is driven by participants' "right to be forgotten" as mandated by regulations such as the General Data Protection Regulation (GDPR)[1] and the California Consumer Privacy Act (CCPA)[2]. While unlearning has been explored in the context of Horizontal Federated Learning (HFL), there has been limited attention to its application in vertical settings. Existing studies on vertical federated unlearning (Zhang et al., 2023a;

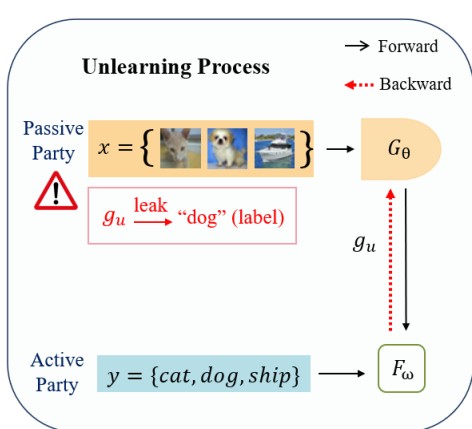

Figure 1: Illustration of the risk of label leakage in vertical federated unlearning (VFU). During VFU, the active party requires to transfer gradient associates with the unlearn features $\mathbf{g}_u$ to the passive party to unlearn the passive model $G_\theta$. As such, this transferred unlearn gradient $\mathbf{g}_u$ poses a potential risk to leak the unlearn label to the passive party. Note that, $F_w$ is active model.

Wang et al., 2024; Deng et al., 2023) primarily focus on the unlearning process for individual clients,

---

[1] https://gdpr-info.eu/art-17-gdpr/
[2] https://oag.ca.gov/privacy/ccpa

often addressing the removal of all features from the passive party upon their exit. In contrast, this paper emphasizes the *unlearning of labels*, which is a critical aspect in VFL, particularly in scenarios such as Credit Risk Assessment where the determination of a loan applicant's likelihood of default is essential. Moreover, the active party aims to eliminate label information not only from the active model but also from the passive models, as the passive models may retain label information (Fu et al., 2022b).

A significant challenge in directly applying traditional machine unlearning methods, such as retraining (Bourtoule et al., 2020; Foster et al., 2023) or Boundary unlearning (Chen et al., 2023), in this context pose a *risk of leaking unlearned labels* during the unlearning process. Typically, the active party, which retains the labels, must either inform the passive party about the samples that require unlearning or transfer the gradients associated with the unlearned label. This practice may inadvertently expose sensitive label information to the passive party (see Fig. 1 and Sect. 3.2).

To address this challenge, we propose a few-shot unlearning method that effectively erases labels from both the active model and passive model in VFL by leveraging a limited amount of private data (see Sect. 4). Specifically, our method employs manifold mixup (Verma et al., 2019) to augment the forward embeddings of each passive party. The active party then performs gradient ascent on the mixed embeddings to unlearn the active model and subsequently transfers the inverse gradients to the passive party to facilitate the unlearning of the passive model independently. This approach offers three key advantages: first, it necessitates only labels from a small amount of private data, significantly reducing the risk of label privacy leakage; second, by utilizing the manifold mixup technique, it enhances unlearning effectiveness with minimal data; and third, it is highly efficient, completing the unlearning process within seconds.

The primary contributions of this work are as follows:

1. To the best knowledge, this is the first work to address the unlearning of labels in VFL.

2. We systematically elucidate the label privacy leakage that may occur when directly applying traditional machine unlearning methods.

3. We propose a few-shot label unlearning method that effectively erases labels from both the active and passive models in VFL, utilizing a limited amount of private data. Moreover, this approach leverages only a small number of data to mitigate the risk of label privacy leakage while employing manifold mixup to enhance unlearning effectiveness.

4. We conduct extensive experiments on multiple benchmark datasets, including MNIST, CIFAR-10, CIFAR-100, and ModelNet, demonstrating that our method rapidly and effectively unlearns target labels compared to other machine unlearning methods.

## 2 RELATED WORKS

**Machine Unlearning & Horizontal Federated Unlearning.** Machine unlearning (MU) was initially introduced by (Cao & Yang, 2015) to selectively remove some data from model without retrain the model from scratch (Garg et al., 2020; Chen et al., 2021). MU can be categorized into exact unlearning and approximate unlearning. Exact unlearning methods such as SISA (Bourtoule et al., 2020) and ARCANE (Yan et al., 2022) split data into sections and train sub-models for each data section and merge all sub-models. During unlearning, retrain the affected data section and merge all sub-models again. In approximate unlearning, techniques such as fine tuning (Golatkar et al., 2020a; Jia et al., 2024) (fine tune with $\mathcal{D}_r$), random label (Graves et al., 2020; Chen et al., 2023) (fine tune with incorrect random label of $\mathcal{D}_u$), noise introducing (Tarun et al., 2024; Huang et al., 2021), gradient ascent (Goel et al., 2023; Choi & Na, 2023; Abbasi et al., 2023; Hoang et al., 2023) (maximise loss associate with $\mathcal{D}_u$), knowledge distillation (Chundawat et al., 2023; Zhang et al., 2023c; Kurmanji et al., 2023) (train a student model) and weights scrubbing (Golatkar et al., 2020a;b; 2021; Guo et al., 2023; Foster et al., 2023) (discarding heavily influenced weights) are used.

Meanwhile, in federated unlearning, most of the existing works are focused in the horizontal environment (Wu et al., 2022; Gu et al., 2024a; Zhao et al., 2024a; Romandini et al., 2024; Liu et al., 2024; Zhang et al., 2023b; Su & Li, 2023; Ye et al., 2023; Gao et al., 2022; Cao et al., 2022; Yuan et al., 2022; Alam et al., 2023; Li et al., 2023; Halimi et al., 2023; Xia et al., 2023; Wang et al., 2023; Dhasade et al., 2023; Liu et al., 2022; Zhao et al., 2024b; Wang et al., 2022; Gu et al., 2024b).

Only very limited research works focus in the vertical environment. For instance, (Zhang et al., 2023a) introduce vertical federated unlearning (VFU) in gradient boosting tree. (Wang et al., 2024) introduce passive party unlearning on deep learning model with fast retraining on remaining parties, and (Deng et al., 2023) introduce passive party unlearning on logistic regression model.

Most if not all existing VFU work have been primarily focused on passive parties unlearning (Zhang et al., 2023a; Wang et al., 2024; Deng et al., 2023). Hence, a significant gap arise when an active party seeks for a collaboration from passive parties for a single class unlearning while all parties remaining engaged in VFL. Unfortunately, current VFU approaches do not address this specific scenario, as they do not explore class unlearning within VFL setting. In contrast to prior works focusing on class unlearning in centralise machine unlearning and horizontal federated unlearning settings, this paper uniquely addresses class unlearning of classification model within the VFL paradigm. This distinction arises because traditional class unlearning methods in centralised and horizontal federated learning setting are impractical for VFL settings, where all parties have different features of data and different computational power.

**Vertical Federated Learning & Privacy Leakage.** VFL is introduced to meet the needs of enterprises looking to utilize features distributed across multiple parties for improved model performance, compared to models trained by a single entity, all while preserving data privacy (Yang et al., 2019). In VFL, privacy is of utmost importance because the participants are typically companies that handle valuable and sensitive user information. Hence, privacy protection during VFU is also an important criteria. We explain the risk of label leakage during VFU in Sect. 3.2.

# 3 LABEL LEAKAGE DURING VERTICAL FEDERATED UNLEARNING

This section explains the risk of label leakage during label unlearning process as depicted in Fig. 1.

## 3.1 GENERAL SETUP

**VFL Training.** We assume that a VFL setting consists of one active party $P_0$ and $K$ passive parties $\{P_1, \cdots, P_K\}$ who collaboratively train a VFL model $\Theta = (\theta, \omega)$ to optimize:

$$
\min_{\omega, \theta_1, \cdots, \theta_K} \frac{1}{n} \sum_{i=1}^{n} \ell(F_\omega \circ (G_{\theta_1}(x_{1,i}), G_{\theta_2}(x_{2,i}), \\
\cdots, G_{\theta_K}(x_{K,i})), y_i),
\tag{1}
$$

in which Party $P_k$ owns features $\mathbf{x}_k = (x_{k,1}, \cdots, x_{k,n})$ and the passive model $G_{\theta_k}$, the active party owns the labels $\mathbf{y} = \{y_1, \cdots, y_m\}$ and active model $F_\omega$. Each passive party $k$ transfers its forward embedding $H_k$ to the active party to compute the loss. The active model $F_\omega$ and passive models $G_{\theta_k}, k \in \{1, \cdots, K\}$ are trained based on backward gradients. Note that, before training, all parties leverage Private Set Intersection (PSI) protocols to align data records with the same IDs. Please see details of the notations in Appendix A.2.

**Unlearning Label in VFL.** When the active party requests to unlearn some sensitive labels $\mathbf{y}^u$, where the corresponding unlearn feature is $\{\mathbf{x}_k^u\}_{k=1}^{K} := \{\{x_{k,i}^u\}_{i=1}^{n_u}\}_{k=1}^{K}$. The active party aims to remove the influence of $\mathbf{y}^u$ on both the active model $F_\omega$ and $K$ passive models $\{G_{\theta_k}\}_{k=1}^{K}$.

Label unlearning in VFL refers to the process of efficiently and securely removing label information from a VFL system. Specifically, the unlearned passive model of client $k$, denoted as $\theta_k^u$, and the unlearned active model, denoted as $\omega^u$, are obtained through the application of an unlearning mechanism $\mathcal{U}$, as follows:
$$
\theta_k^u = \mathcal{U}(\theta_k, \mathbf{g}_u), \quad \omega^u = \mathcal{U}(\omega, \mathbf{y}_u),
$$
where $\theta_k$ and $\omega$ represent the passive models of client $k$ and active model before unlearning, respectively, and $\mathbf{g}_u$ are the gradients associated with the unlearned label $\mathbf{y}_u$.

Building upon the principles of machine unlearning presented in (Bourtoule et al., 2020), label unlearning in VFL needs to satisfy the following three objectives: i) **Selective Removal**: The influence of specific labels must be erased while preserving the integrity of other data. ii) **Efficiency**: The unlearning process should achieve the above without requiring the computational cost of retraining

the model from scratch. iii) **Privacy Preservation**: The unlearning process must ensure that no sensitive label information is leaked to the passive party.

**Threat Model.** We assume all participating parties are *semi-honest* and do not collude with each other. An adversary (i.e., the passive party) faithfully executes the training protocol but may launch privacy attacks to infer the private labels of the active party.

**Assumption.** We assume that the passive party possesses corresponding labels for a limited number of features, defined as $\mathcal{D}^p = \{(\mathbf{x}_k^p, \mathbf{y}^p)\}_{k=1}^K = \{\{(x_{k,i}^p, y_i)\}_{i=1}^{n_p}\}_{k=1}^K$, where $n_p << n_u$. This assumption is reasonable, as the active party must convey some label information to the passive party in order to effectively remove that information. Furthermore, this assumption is widely employed in prior works (Fu et al., 2022b; Gu et al., 2023; Zou et al., 2022).

### 3.2 LABEL LEAKAGE DURING UNLEARNING

To remove the influence of the passive models $\{G_{\theta_k}\}_{k=1}^K$, there exists a risk of unlearning label leakage ($\mathbf{y}_u = \{y_1^u, \ldots, y_{m_u}^u\}$) to the passive parties. During the unlearning process, the active party is required to transfer information to the passive party, e.g., gradients $\mathbf{g}_u = \{g_1^u, \ldots, g_{n_u}^u\}$, in order to effectively unlearn the label associated with the passive model. Consequently, the passive party may infer the label based on this information.

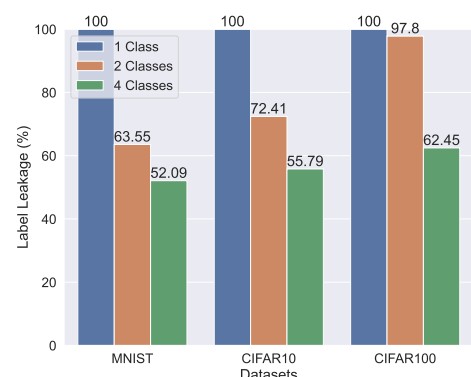

Figure 2: Illustration of label leakage (%) with Boundary unlearning in VFL using ResNet18 model on different number of classes and datasets.

In particular, when unlearning a single class $y_{u,1}$, we consider two representative unlearning methods: (i) retraining (Foster et al., 2023) and (ii) Boundary unlearning (Chen et al., 2023). For retraining methods, the active party must inform the passive party regarding which features do not require training, thus, the label is leaked. In the case of Boundary unlearning, the gradients transferred to the passive party correspond to the features associated with the label $y_{u,1}$ may leak the label.

Furthermore, when multiple labels ($m_u$) are targeted for unlearning, the label leakage issue becomes exacerbated. Lets consider the Boundary unlearning as an example. This method illustrates that the passive party can infer label information from the gradients $\mathbf{g}_u$ transmitted by the active party during the unlearning process. Specifically, the passive party employs clustering on $\mathbf{g}_u$ to derive $m_u$ clusters by optimizing the following objective function:

$$\min \sum_{g_i \in \mathcal{C}_j} \sum_{j=1}^{m_u} |g_{u,i} - \bar{g}_{u,j}|, \tag{2}$$

where $\mathcal{C}_j$ denotes the set of points assigned to cluster $j$, and $\bar{g}_{u,j}$ represents the centroid of cluster $j$. Consequently, the passive party can deduce the labels of the features in $\mathcal{X}$. Fig. 2 exposes the label leakage (in %) during unlearning in VFL for varying numbers of unlearning classes. For instance, with four classes from CIFAR-100, a total of 62.45% of label leakage is exposed.

## 4 THE PROPOSED FEW-SHOT LABEL UNLEARNING METHOD

This section details the proposed few-shot label unlearning method as illustrated in Fig. 3 and Algorithm 1. Our solution comprises two primary steps: first, augmenting the forward embedding through manifold mixup to address the scarcity of labeled data for unlearning (see Sect. 4.1). Second, employing gradient ascent on the augmented embedding to influence both the passive and active models, thereby facilitating the removal of the specified class, as elaborated in Sect. 4.2.

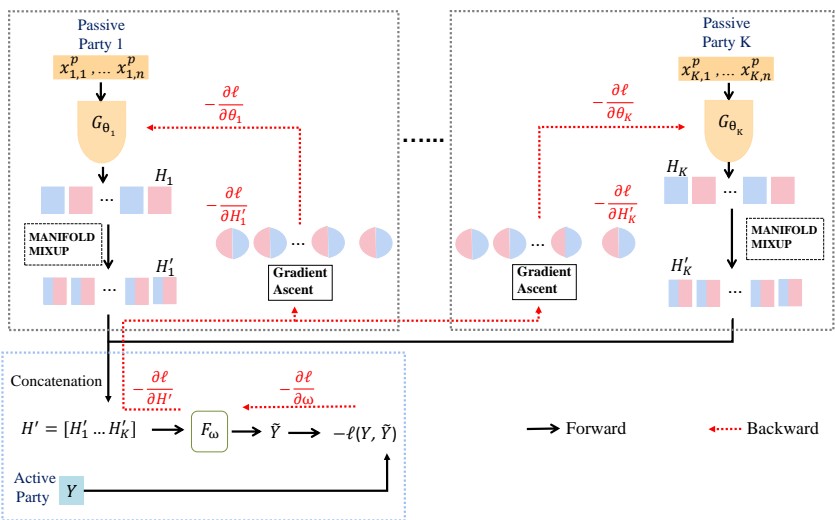

Figure 3: Overview of our proposed few-shot unlearning framework in VFL setting.

## 4.1 VERTICAL MANIFOLD MIXUP

Due to the label privacy leakage issue (Sect. 3.2), directly applying traditional machine unlearning methods will pose some challenges. We assume that the active party discloses a limited number of labels to the passive party to facilitate the unlearning of a specific class. However, this small labeled dataset, denoted as $\mathcal{D}_p = \{(x^p_{1,i}, x^p_{2,i}, \cdots, x^p_{K,i}, y^p_i)\}_{i=1}^{n_p}$, is insufficient for an effective unlearning (see Appendix). Consequently, this scenario can be framed as a few-shot unlearning problem, wherein a minimal set of labels is employed to unlearn all associated labels.

Drawing inspiration from the few-shot learning principles, we adopt the manifold mixup mechanism (Verma et al., 2019) by interpolating hidden embeddings rather than directly mixing the features. We propose a manifold mixup framework for VFL by optimizing the following loss function:

$$\min_{\omega,\theta_1,\cdots,\theta_K} \frac{1}{n_p^2} \sum_{i,j=1}^{n_p} \ell(F_\omega \circ (\text{Mix}_\lambda(G_{\theta_1}(x^p_{1,i}), G_{\theta_1}(x^p_{1,j})),$$

$$\cdots, \text{Mix}_\lambda(G_{\theta_K}(x^p_{K,i}), G_{\theta_K}(x^p_{K,j})), \text{Mix}_\lambda(y^p_i, y^p_j)),$$

where

$$\text{Mix}_\lambda(a, b) = \lambda \cdot a + (1 - \lambda) \cdot b. \quad (3)$$

The mixed coefficient $\lambda$ ranges from 0 to 1. The advantage of the manifold mixup approach lies in its ability to flatten the state distributions (Verma et al., 2019). Specifically, for each passive party $k$, mixup is applied to the forward embeddings $\{H^p_k = G_\theta(x^p_{k,i})\}$ to generate numerous mixed embeddings $H'_k$. Subsequently, all passive parties transfer their respective mixed embeddings $H'_k$ to the active party.

**Algorithm 1** Our Method

**Input:** Bottom models parameters $\theta_k$ of $K$ passive parties, top model parameters $\omega$, unlearn data $\mathcal{D}_u$, learning rate $\eta$, unlearn epoch $N$.

**Output:** Unlearned bottom models parameters $\theta_k^u$, unlearned top model parameters $\omega^u$

1: Initialize model $\theta_k^u$ and $\omega^u$ before unlearning
2: **for** $n$ in $N$ **do**:
3:     **for** $(x^p_i, y^p_i)$ in $\mathcal{D}_p$ **do**:
4:       ▷ *Passive parties $k$:*
5:       Split $x^p_i$ to $K$ parts.
6:       **for** $k = 1$ to $K$ **do**:
7:         $H^p_k = G_{\theta_k}(x^p_{k,i})$
8:         Generate $H'_k$ from $H_k$ according to equation 3.
9:       ▷ *Active party:*
10:       $H' = [H'_1, ..., H'_K]$
11:       $y = F_\omega(H')$.
12:       $L = \ell(y, y')$
13:       $\omega = \omega + \eta \cdot \frac{\partial L}{\partial \omega}$
14:       Active party compute $\frac{\partial \ell}{\partial H'_k}$ to transfer all passive parties.
15:       ▷ *Passive parties $k$:*
16:       **for** $k = 1$ to $K$ **do**:
17:         $g_k = \frac{\partial \ell}{\partial H'_k} \cdot \frac{\partial H_k}{\partial \omega}$
18:         $\theta_k = \theta_k + \eta \cdot g_k$
**Return** $\theta_k^u$ and $\omega^u$.

## 4.2 VERTICAL LABEL UNLEARNING VIA GRADIENT ASCENT

Once the augmented embeddings $\{H'_1, \ldots, H'_K\}$ for the representative unlearned data $\mathcal{D}_p$ (label is known) are generated, a straightforward yet effective strategy is to implement gradient ascent for both the active and passive models using these augmented embeddings. Specifically, the active party concatenates all embeddings $\{H'_k\}_{k=1}^K$ into a single tensor $H' = [H'_1, \ldots, H'_K]$, and optimizes it according to the following formulation:

$$\min_\omega \ell(F_\omega(H'), y') = \ell(F_\omega([H'_1, \ldots, H'_K]), y'), \tag{4}$$

where $y'$ represents the mixture of the representative unlearned labels and $\eta$ is the learning rate.

**Unlearning for active model** $F_\omega$. On one hand, the active model undergoes unlearning for active model $F_\omega$ via gradient ascent as follows:

$$\omega = \omega + \eta \nabla_\omega \ell(F_\omega(H'), y'). \tag{5}$$

**Unlearning for passive model** $G_{\omega_k}$. Subsequently, the active party computes the gradients $g'_k = \frac{\partial \ell}{\partial H'_k}$ in accordance with equation 4 and transmits these gradients to the corresponding passive party $k$. Finally, the passive party $k$ updates the passive model $G_{\theta_k}$ using the following expression:

$$\theta_k = \theta_k + \eta \nabla_{H'_k} \ell(F_\omega(H'), y') \cdot \nabla_{\theta_k} H'_k. \tag{6}$$

It is important to note that gradient ascent may lead to significant degradation in model utility or even result in vanishing gradients if the parameters are not appropriately tuned. Therefore, employing a small learning rate $\eta$ and a limited number of unlearning epochs can mitigate these issues while achieving effective unlearning results (see discussion in Appendix A.1 and experimental details in Appendix A.4).

## 5 EXPERIMENTAL RESULTS

This section presents the empirical analysis of the proposed method in terms of utility, unlearning effectiveness, time efficiency and some ablation studies.

### 5.1 EXPERIMENT SETUP

**VFL Setting.** We stimulate a VFL scenario including one active party owned the active model and multiple passive parties (ranges from one(1) to eight(8)) owned the passive model (see more details in Appendix A.3).

**Datasets & Models.** We conduct experiments on six datasets: MNIST (Lecun et al., 1998), CIFAR10, CIFAR100 (Krizhevsky et al., 2009), ModelNet (Wu et al., 2015), Brain Tumor MRI (Wang et al., 2024) and Yahoo Answers dataset (Fu et al., 2022a). We adopt ResNet18 (He et al., 2015) on dataset MNIST, CIFAR10, CIFAR100, ModelNet and Brain Tumor MRI. We adopt MixText (Chen et al., 2020) on Yahoo Answers dataset. We do extend our experiments with Vgg16 (Simonyan & Zisserman, 2015) on dataset CIFAR10 and CIFAR100. Experiments are repeated over five random trials, and results are reported as mean and standard deviation. Experiment results on Brain Tumor MRI and Yahoo Answer datasets and further details are available in Appendix A.3. For the MNIST, CIFAR10, and CIFAR100 datasets, each image feature is divided among $K$ parties, where $K$ represents the number of passive parties. For the ModelNet dataset, we generate $K$ 2D multi-view images per 3D mesh model by placing two virtual cameras evenly distributed around the centroid. Each passive party is assigned one of the $K$ generated 2D multi-view images.

**Evaluations Metrics.** We evaluate the utility of unlearning by measuring accuracy of $\mathcal{D}_r$ before and after unlearning. The higher accuracy on $\mathcal{D}_r$ indicates stronger utility. To evaluate the unlearning effectiveness, we construct a simple MIA from (Shokri et al., 2017) to test Attack Success Rate (ASR) and measuring the accuracy of $\mathcal{D}_u$ before and after unlearning. MIA seeks to determine if a specific data record was included in the training of a target machine learning model. Time efficiency is evaluated by comparing the runtime of each baseline.

| Model | Datasets | Metrics | Accuracy (%) | | | | | | | |
|---|---|---|---|---|---|---|---|---|---|---|
| | | | Baseline | Retrain | FT | Fisher | Amnesiac | Unsir | BU | Ours |
| ResNet18 | MNIST | $\mathcal{D}_r$ | 99.29 | 99.33 ± 0.03 | **98.99 ± 0.05** | 12.16 ± 0.46 | 98.16 ± 0.92 | 84.92 ± 1.13 | 98.72 ± 0.02 | 98.89 ± 0.00 |
| | | $\mathcal{D}_u$ | 99.39 | 0.00 ± 0.00 | **0.00 ± 0.00** | 0.00 ± 0.00 | 0.00 ± 0.00 | 0.00 ± 0.00 | 58.83 ± 1.79 | **0.00 ± 0.00** |
| | | ASR | 90.61 | 1.03 ± 0.24 | 2.92 ± 1.08 | **0.11 ± 0.07** | 0.00 ± 0.00 | 29.07 ± 7.95 | **0.47 ± 0.01** | **0.63 ± 0.01** |
| | CIFAR10 | $\mathcal{D}_r$ | 90.61 | 91.26 ± 0.12 | 88.16 ± 0.15 | 54.4 ± 10.77 | 86.37 ± 0.20 | 75.02 ± 1.65 | 72.65 ± 0.55 | **89.11 ± 0.14** |
| | | $\mathcal{D}_u$ | 93.10 | 0.00 ± 0.00 | 11.00 ± 0.10 | 0.00 ± 0.00 | 0.00 ± 0.00 | 0.00 ± 0.00 | 3.25 ± 0.15 | **0.00 ± 0.00** |
| | | ASR | 83.84 | 25.98 ± 1.27 | **15.85 ± 2.33** | 50.67 ± 12.51 | **1.62 ± 0.54** | 76.78 ± 0.44 | 34.90 ± 1.16 | **18.21 ± 0.63** |
| | CIFAR100 | $\mathcal{D}_r$ | 71.43 | 71.03 ± 0.12 | 66.86 ± 0.73 | 61.04 ± 8.61 | 60.05 ± 0.03 | 59.32 ± 0.14 | 55.30 ± 0.81 | **67.85 ± 0.03** |
| | | $\mathcal{D}_u$ | 83.00 | 0.00 ± 0.00 | 12.25 ± 2.25 | 0.00 ± 0.00 | 0.00 ± 0.00 | 0.00 ± 0.00 | 3.50 ± 0.50 | **0.00 ± 0.00** |
| | | ASR | 88.40 | 25.53 ± 3.36 | 29.30 ± 2.70 | 28.10 ± 4.10 | **2.60 ± 1.30** | 73.70 ± 1.70 | **6.00 ± 0.60** | **13.47 ± 0.19** |
| | ModelNet | $\mathcal{D}_r$ | 94.26 | 93.90 ± 0.11 | 66.64 ± 1.53 | 28.10 ± 0.69 | 73.91 ± 1.83 | 13.51 ± 0.05 | 24.07 ± 0.27 | **83.32 ± 0.07** |
| | | $\mathcal{D}_u$ | 100.00 | 0.00 ± 0.00 | **0.00 ± 0.00** | 0.00 ± 0.00 | 0.00 ± 0.00 | 0.00 ± 0.00 | 0.00 ± 0.00 | 2.00 ± 0.00 |
| | | ASR | 98.40 | 0.65 ± 0.05 | 0.79 ± 0.16 | 23.48 ± 0.77 | 1.11 ± 0.16 | 49.20 ± 1.25 | 21.16 ± 0.23 | **0.46 ± 0.07** |
| Vgg16 | CIFAR10 | $\mathcal{D}_r$ | 89.50 | 90.27 ± 0.19 | 88.69 ± 0.08 | 15.93 ± 4.82 | 84.67 ± 0.22 | 74.74 ± 0.72 | 82.69 ± 0.1 | **88.85 ± 0.24** |
| | | $\mathcal{D}_u$ | 91.10 | 0.00 ± 0.00 | 4.25 ± 1.05 | 0.00 ± 0.00 | 0.00 ± 0.00 | 0.00 ± 0.00 | 2.85 ± 0.05 | 1.60 ± 0.16 |
| | | ASR | 81.66 | 33.10 ± 1.86 | **21.84 ± 2.66** | 42.25 ± 6.23 | **2.36 ± 0.86** | 21.75 ± 2.41 | 34.53 ± 0.65 | **31.59 ± 0.34** |
| | CIFAR100 | $\mathcal{D}_r$ | 65.48 | 65.32 ± 0.32 | 59.92 ± 0.56 | 35.42 ± 1.95 | 55.83 ± 0.13 | 55.78 ± 0.59 | 52.21 ± 0.00 | **62.13 ± 0.06** |
| | | $\mathcal{D}_u$ | 77.00 | 0.00 ± 0.00 | 2.50 ± 0.25 | 0.00 ± 0.00 | 0.00 ± 0.00 | 0.00 ± 0.00 | 3.00 ± 0.00 | 4.30 ± 0.94 |
| | | ASR | 87.20 | 42.13 ± 2.73 | **34.50 ± 4.30** | 40.70 ± 3.50 | **3.10 ± 0.15** | 42.70 ± 0.70 | **18.20 ± 0.11** | **21.73 ± 0.84** |

Table 1: Accuracy of $\mathcal{D}_r$ and $\mathcal{D}_u$ for each unlearning method across ResNet18 and Vgg16 model in single-class unlearning

**Unlearning Scenarios.** *Single-class unlearning*: We forget a single class from all datasets. *Two-classes unlearning*: We forget two classes from CIFAR10/100. *Multi-classes unlearning*: We forget four classes from CIFAR100. Note that, the labels selected for unlearning remain consistent across all datasets. Specifically: a) In single-label unlearning, we unlearn label *"0"*; b) In two-label unlearning, we unlearn labels *"0"* and *"2"*, respectively. While, c) In multi-label unlearning, we unlearn labels *"0"*, *"2"*, *"5"*, and *"7"*, respectively.

**Baselines.** We compare our method with the following baselines: Retrain, Fine Tuning (Golatkar et al., 2020a; Jia et al., 2024), Fisher Forgetting (Golatkar et al., 2020a), Amnesiac Unlearning (Graves et al., 2020), UNSIR (Tarun et al., 2024) and Boundary Unlearning (Chen et al., 2023). We implement the baselines with the following details. *Retrain*: Retrain the model from scratch with $\mathcal{D}_r$ with the same hyper-parameters to baseline. *Fine Tuning* (Golatkar et al., 2020a; Jia et al., 2024): The baseline model is fine-tuned using $\mathcal{D}_r$ for 5 epochs with learning rate set 0.01. *Fisher Forgetting* (Golatkar et al., 2020a): We use fisher information matrix (FIM) to inject noise into the parameters proportional to their relative importance to the $\mathcal{D}_f$ compared to the $\mathcal{D}_r$. *Amnesiac* (Graves et al., 2020): We retrain the model for 3 epochs with relabeled $\mathcal{D}_f$ with incorrect random label and $\mathcal{D}_r$. *Unsir* (Tarun et al., 2024): We introduce noise matrix on $\mathcal{D}_f$ to impair the model with noise generated and repair the model with $\mathcal{D}_r$. *Boundary Unlearning* (Chen et al., 2023): We create adversarial examples from $\mathcal{D}_f$ and assign new nearest incorrect adversarial label to shrink the $\mathcal{D}_f$ to the nearest incorrect decision boundary.

## 5.2 EXPERIMENTAL RESULTS

### 5.2.1 UTILITY GUARANTEE

To assess the utility of our proposed unlearning method, we evaluate accuracy on $\mathcal{D}_r$ before and after unlearning (Tab. 1, 2, 3). An effective unlearning method should retain as much information as possible from $\mathcal{D}_r$.

From Tab. 1, 2, 3, we observe that: i) Fine-tuning achieves good preservation on $\mathcal{D}_r$, but its unlearning effectiveness is low (see Sect. 5.2.2). ii) Fisher forgetting badly preserves the information of $\mathcal{D}_r$, resulting in a huge degradation on $\mathcal{D}_r$ accuracy. iii) Random incorrect labeling of $\mathcal{D}_u$ from Amnesiac Unlearning causes the decision boundaries of $\mathcal{D}_r$ to shift unpredictably, resulting in a drop in accuracy on $\mathcal{D}_r$. This degradation is more pronounced in datasets with a large number of classes, such as CIFAR100 and ModelNet. iv) The repair step from UNSIR fails to fully retain the information in $\mathcal{D}_r$, leading to some performances degradation on $\mathcal{D}_r$. v) Boundary unlearning

| Model | Datasets | Metrics | Accuracy (%) | | | | | | | |
| | | | Baseline | Retrain | FT | Fisher | Amnesiac | Unsir | BU | Ours |
|---|---|---|---|---|---|---|---|---|---|---|
| ResNet18 | CIFAR10 | $\mathcal{D}_r$ | 91.48 | 91.74 ± 0.01 | **90.63 ± 0.57** | 31.25 ± 2.23 | 86.16 ± 0.82 | 74.48 ± 0.06 | 81.64 ± 0.56 | 88.25 ± 0.09 |
| | | $\mathcal{D}_u$ | 88.40 | 0.00 ± 0.00 | 41.15 ± 1.55 | 49.55 ± 0.40 | **0.00 ± 0.00** | **0.00 ± 0.00** | 19.90 ± 0.85 | 0.63 ± 0.60 |
| | | ASR | 79.61 | 21.66 ± 0.64 | **13.22 ± 0.37** | 25.60 ± 0.08 | **1.84 ± 0.13** | 41.79 ± 1.35 | 35.40 ± 1.54 | 28.20 ± 1.48 |
| | CIFAR100 | $\mathcal{D}_r$ | 71.56 | 71.21 ± 0.13 | 66.04 ± 0.58 | 53.56 ± 2.54 | 59.52 ± 0.03 | 58.02 ± 0.37 | 56.37 ± 0.39 | **66.89 ± 0.05** |
| | | $\mathcal{D}_u$ | 71.00 | 0.00 ± 0.00 | 38.00 ± 0.01 | 25.20 ± 5.75 | **0.00 ± 0.00** | **0.00 ± 0.00** | 13.00 ± 0.01 | 6.50 ± 0.71 |
| | | ASR | 88.60 | 21.60 ± 0.85 | 19.20 ± 1.20 | 48.90 ± 0.54 | **6.50 ± 0.40** | 54.83 ± 0.44 | **13.70 ± 0.90** | 6.50 ± 0.33 |
| Vgg16 | CIFAR10 | $\mathcal{D}_r$ | 89.80 | 91.13 ± 0.03 | 88.09 ± 0.35 | 47.53 ± 2.38 | 86.16 ± 0.19 | 71.50 ± 0.07 | 88.67 ± 0.22 | **88.21 ± 0.02** |
| | | $\mathcal{D}_u$ | 89.10 | 0.00 ± 0.00 | 28.55 ± 0.33 | 13.10 ± 0.28 | **0.00 ± 0.00** | **0.00 ± 0.00** | 19.08 ± 0.53 | **0.00 ± 0.00** |
| | | ASR | 82.64 | 28.31 ± 1.23 | **17.75 ± 2.22** | 68.43 ± 1.14 | **1.67 ± 0.01** | 46.21 ± 0.72 | **11.72 ± 0.07** | 28.37 ± 0.86 |
| | CIFAR100 | $\mathcal{D}_r$ | 65.75 | 65.59 ± 0.17 | 60.79 ± 0.37 | 35.24 ± 2.21 | 57.86 ± 0.81 | 56.04 ± 0.44 | 50.02 ± 0.18 | **62.49 ± 0.11** |
| | | $\mathcal{D}_u$ | 58.50 | 0.00 ± 0.00 | 11.75 ± 1.25 | 11.00 ± 4.85 | **0.00 ± 0.00** | **0.00 ± 0.00** | 3.25 ± 0.25 | **0.00 ± 0.00** |
| | | ASR | 73.60 | 30.55 ± 0.05 | **22.75 ± 1.05** | 32.60 ± 1.17 | **3.45 ± 0.65** | 52.40 ± 0.80 | 27.90 ± 1.20 | 30.50 ± 1.80 |

Table 2: Accuracy of $\mathcal{D}_r$ and $\mathcal{D}_u$ for each unlearning method across ResNet18 and Vgg16 model in two-classes unlearning

exhibits inconsistencies across different datasets, models, and scenarios. In some cases, they show huge degradation on $\mathcal{D}_r$, while in other instances, they preserve $\mathcal{D}_r$ well. Contrary, vi) our solution shows good unlearning utility in all experiment settings.

### 5.2.2 UNLEARNING EFFECTIVENESS

For unlearning effectiveness, we run MIA to evaluate if the unlearned model leaks any information about the $\mathcal{D}_u$ and measure the accuracy of $\mathcal{D}_u$ before and after unlearning.

From Tab. 1, 2, 3, we observe that: i) Fine-tuning shows bad unlearning effectiveness on CIFAR10/100 datasets. The unlearning effectiveness of fine tuning is worse on two-classes (Tab. 2) and multi-classes unlearning scenarios (Tab. 3); ii) Fisher forgetting, Amnesiac Unlearning and UNSIR show strong unlearning effectiveness, reducing accuracy of $\mathcal{D}_u$ to 0.00%; iii) Boundary unlearning exhibits inconsistencies across different datasets, models, and scenarios. In some cases, they show good unlearning effectiveness on $\mathcal{D}_u$, while in other instances, they show bad unlearning effectiveness. In contrast, iv) our solution demonstrates strong effectiveness across all models, datasets, and scenarios. It achieves successful unlearning of $\mathcal{D}_u$.

Also, on the same tables (Tab. 1-3), we observe that: i) Fine tuning shows consistent ASR score. ii) Fisher forgetting shows high ASR score in most of the cases. iii) Amnesiac unlearning shows consistencies in very low ASR score across all experiments. iv) UNSIR shows high ASR score on almost all experiments, v) Boundary unlearning shows relatively consistent ASR scores. Finally, all in all vi) our solution shows a consistent ASR performance across all datasets, models and scenarios.

### 5.2.3 TIME EFFICIENCY

For the computational complexity, Fig. 4 presents an execution time (in seconds) of single-class unlearning with ResNet18 model in CIFAR10 dataset. It can be observed that: i) The gold standard retrain model has the highest execution time. ii) Unlearning methods that utilises full dataset or $\mathcal{D}_r$ such as Fine Tuning, Amnesiac Unlearning and Fisher forgetting have relatively high execution time. iii) Unlearning methods that utilise only $\mathcal{D}_u$ such as Boundary Unlearning shows a lower execution time. iv) Our solution has the lowest execution time (16x - 1200x lower).

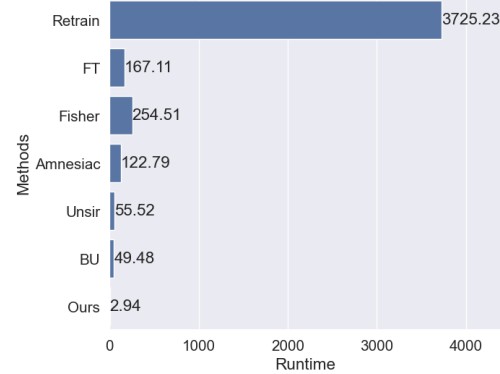

Figure 4: The runtime(s) of each unlearning method.

| Model | Datasets | Metrics | Accuracy (%) | | | | | | | |
|---|---|---|---|---|---|---|---|---|---|---|
| | | | Baseline | Retrain | FT | Fisher | Amnesiac | Unsir | BU | Ours |
| ResNet18 | CIFAR100 | $\mathcal{D}_r$ | 71.53 | 71.91 ± 0.12 | 67.16 ± 0.13 | 54.79 ± 1.04 | 59.09 ± 0.54 | 59.05 ± 0.38 | 48.96 ± 0.04 | **69.87 ± 0.09** |
| | | $\mathcal{D}_u$ | 72.00 | 0.00 ± 0.00 | 33.87 ± 0.88 | 45.38 ± 1.13 | **0.00 ± 0.00** | **0.00 ± 0.00** | 15.00 ± 0.25 | 4.83 ± 1.12 |
| | | ASR | 86.65 | 16.95 ± 0.35 | 18.23 ± 1.63 | 62.78 ± 3.93 | **6.05 ± 1.19** | 68.63 ± 1.83 | 38.35 ± 0.75 | **13.97 ± 0.45** |
| Vgg16 | CIFAR100 | $\mathcal{D}_r$ | 65.83 | 65.66 ± 0.08 | 60.92 ± 0.08 | 36.55 ± 1.07 | 57.26 ± 0.18 | 56.86 ± 0.26 | 47.04 ± 0.32 | **64.33 ± 0.16** |
| | | $\mathcal{D}_u$ | 60.25 | 0.00 ± 0.00 | 7.63 ± 0.13 | 28.75 ± 1.25 | **0.00 ± 0.00** | **0.00 ± 0.00** | 7.13 ± 0.11 | 6.00 ± 0.25 |
| | | ASR | 75.80 | 27.20 ± 0.75 | **24.38 ± 3.13** | 55.20 ± 3.75 | **4.80 ± 0.05** | 32.83 ± 0.58 | 29.70 ± 0.03 | 27.50 ± 0.65 |

Table 3: Accuracy of $\mathcal{D}_r$ and $\mathcal{D}_u$ for each unlearning method across ResNet18 and Vgg16 model in multi-classes unlearning

| Number of Passive Parties | Metrics | Accuracy (%) | | | | | | | |
|---|---|---|---|---|---|---|---|---|---|
| | | Baseline | Retrain | FT | Fisher | Amnesiac | Unsir | BU | Ours |
| 1 | $\mathcal{D}_r$ | 92.50 | 93.27 ± 0.11 | 88.51 ± 0.09 | 76.83 ± 3.02 | 88.95 ± 0.58 | 77.89 ± 0.48 | 89.66 ± 0.08 | **90.01 ± 0.46** |
| | $\mathcal{D}_u$ | 93.60 | 0.00 ± 0.00 | **0.00 ± 0.00** | **0.00 ± 0.00** | **0.00 ± 0.00** | **0.00 ± 0.00** | 23.60 ± 1.60 | **0.00 ± 0.00** |
| | ASR | 89.34 | 24.54 ± 1.38 | 40.27 ± 3.15 | 66.40 ± 1.98 | **0.36 ± 0.14** | 15.83 ± 0.49 | 19.66 ± 0.56 | 16.13 ± 0.36 |
| 2 | $\mathcal{D}_r$ | 90.61 | 91.26 ± 0.12 | 88.16 ± 0.15 | 54.40 ± 10.77 | 86.37 ± 0.20 | 75.02 ± 1.65 | 72.65 ± 0.55 | **89.11 ± 0.14** |
| | $\mathcal{D}_u$ | 93.10 | 0.00 ± 0.00 | 11.00 ± 0.10 | **0.00 ± 0.00** | **0.00 ± 0.00** | **0.00 ± 0.00** | 3.25 ± 0.15 | **0.00 ± 0.00** |
| | ASR | 83.84 | 25.98 ± 1.27 | **15.85 ± 2.33** | 50.67 ± 12.51 | **1.62 ± 0.54** | 76.78 ± 0.44 | 34.90 ± 1.16 | 18.21 ± 0.63 |
| 4 | $\mathcal{D}_r$ | 88.12 | 89.04 ± 0.02 | 77.52 ± 1.15 | 41.56 ± 0.49 | 81.77 ± 0.04 | 71.88 ± 0.39 | 73.85 ± 0.49 | **86.69 ± 0.13** |
| | $\mathcal{D}_u$ | 91.40 | 0.00 ± 0.00 | **0.00 ± 0.00** | 0.90 ± 0.00 | **0.00 ± 0.00** | **0.00 ± 0.00** | 1.81 ± 0.03 | **0.00 ± 0.00** |
| | ASR | 79.58 | 25.86 ± 2.04 | 63.44 ± 0.44 | 52.05 ± 0.91 | **2.90 ± 0.38** | 76.52 ± 4.16 | 72.61 ± 0.97 | 21.51 ± 0.69 |

Table 4: Accuracy of $\mathcal{D}_r$ and $\mathcal{D}_u$ for each unlearning method across ResNet18 model in single-class unlearning on different number of passive parties.

## 5.3 ABLATION STUDY

In this section, we conduct an ablation study on the effectiveness of our method for different number of passive parties and different privacy-preserving VFL mechanishm.

### 5.3.1 EVALUATION ON DIFFERENT SIZE OF $D_p$

We apply the gradient ascent with different size $D_p$ to achieve unlearning in Fig. 5, e.g, three methods (GA-A using 5000 samples, GA-S using 40 samples and ours). It shows that i) 40 samples is not enough to unlearn since the unlearning result on $D_u$ remains at 40.48% while GA-A with 5000 samples achieves 0%. Meanwhile, ii) our method with only 40 samples able to achieve 0% unlearning effectiveness on $D_u$ (see more experiment in Appendix A.4).

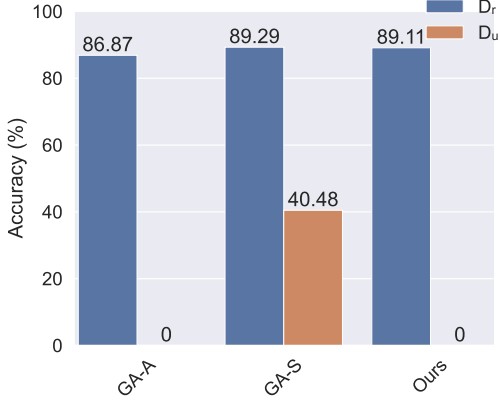

Figure 5: Comparison of the utility and unlearning effectiveness on different size of $D_p$. The results indicate that when using a limited amount of data ($|D_p| = 40$), directly applying gradient ascent (GA-S) does not achieve satisfactory unlearning effectiveness, as the accuracy on the unlearned data remains at 40.48%. Contrary, our method, which incorporates manifold mixup, demonstrates significantly better unlearning effectiveness (e.g. with only 40 labeled data points, our approach reduces the unlearned accuracy to 0%.)

### 5.3.2 EVALUATION FOR DIFFERENT NUMBER OF PASSIVE PARTIES

Table 4 shows the accuracy of $\mathcal{D}_r$, $\mathcal{D}_u$ and ASR score on one(1) passive party, two(2) and four(4) passive parties, respectively. The results indicate that our method can perform well in unlearning effectiveness and utility.

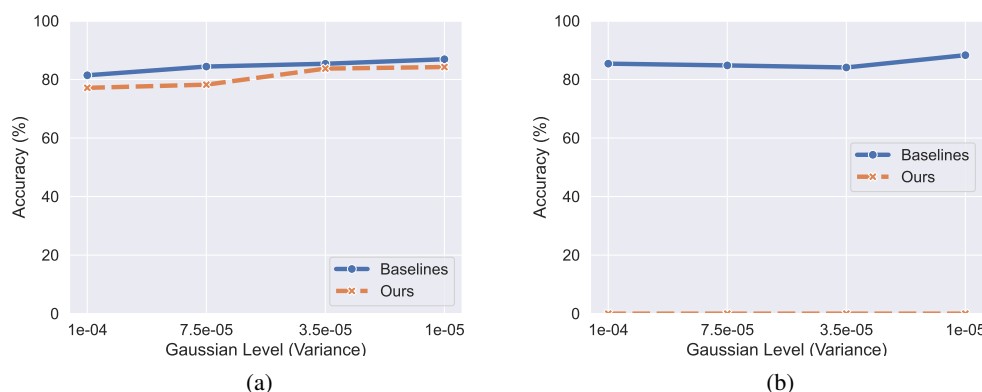

Figure 6: Comparison of the utility and unlearning effectiveness for Differential Privacy (Fu et al., 2022b) (a privacy preserving VFL method). (a) and (b) show the accuracy of $\mathcal{D}_r$ and $\mathcal{D}_u$ between baseline and our solution on different level of Gaussian Noise model, respectively.

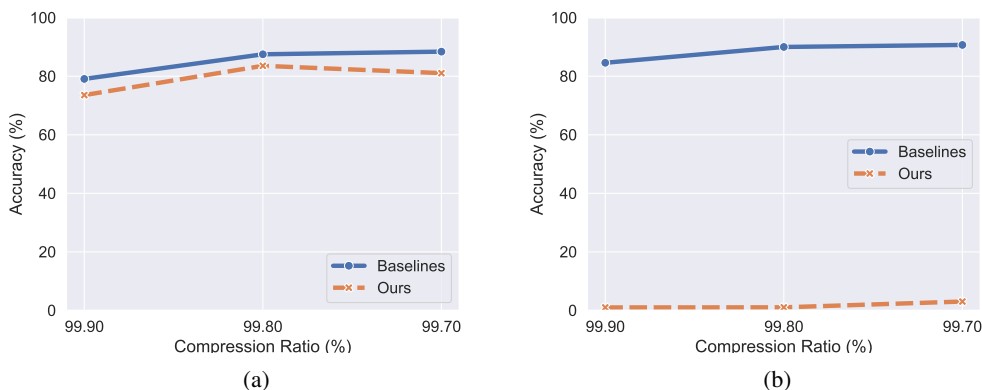

Figure 7: Comparison of the utility and unlearning effectiveness for Gradient Compression (Fu et al., 2022b) (a privacy preserving VFL method). (a) and (b) show the accuracy of $\mathcal{D}_r$ and $\mathcal{D}_u$ between baseline and our solution on different level of gradient compression ratio model, respectively.

### 5.3.3 EVALUATION FOR DIFFERENT PRIVACY PRESERVING VFL METHODS

We evaluate our unlearning methods under two privacy preserving VFL methods: (i) Differential Privacy (Fu et al., 2022b) and (ii) Gradient Compression (Fu et al., 2022b). Fig. 6 and 7 present the effectiveness of our solution on both methods across different levels of variance Gaussian noise and compression ratio, respectively. It shows that even for a large compression ratio and noise level, our proposed method still able to unlearn effectively, while the utility of the vertical training decreases significantly.

## 6 CONCLUSIONS

In conclusion, this paper presents a pioneering approach to label unlearning within VFL domain, addressing a critical gap in the existing literature. By introducing a few-shot unlearning method that leverages manifold mixup, we effectively mitigate the risk of label privacy leakage while ensuring efficient unlearning from both active and passive models. Our systematic exploration of potential label privacy risks and extensive experimental validation on benchmark datasets underscores the proposed method's efficacy and rapid performance. Ultimately, this work not only advances the understanding of unlearning in VFL but also sets the stage for further innovations in privacy-preserving collaborative machine learning practices.

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

# A   APPENDIX

This section provides a detailed information on discussion, our experimental settings and additional experimental results.

## A.1   DISCUSSION FOR UNLEARNING EFFECTIVENESS

Consider a scenario where the active party seeks to unlearn the label $y_u$ with the corresponding feature $x_u$ and embedding $H_u = G_\theta(x_u)$. The gradient ascent approach aims to remove the label information $y_u$ from both the active model $\theta$ and the passive model $\omega$.

1) **Unlearning effectiveness for Gradient Ascent (GA).** Using the first-order Taylor expansion of $\ell(\omega; H_u, y_u)$ around the initial parameter $\omega_t$, we obtain:

$$\ell(\omega_{t+1}; H_u, y_u) \approx \ell(\omega_t; H_u, y_u) + \nabla_\omega \ell(\omega_t; H_u, y_u)^\top (\omega_{t+1} - \omega_t).$$

Substituting the gradient ascent update $\omega_{t+1} = \omega_t + \eta \nabla_\omega \ell(\omega_t; H_u, y_u)$ (as defined in Eq. (5) of the main text), this becomes:

$$\ell(\omega_{t+1}; H_u, y_u) \approx \ell(\omega_t; H_u, y_u) + \eta \|\nabla_\omega \ell(\omega_t; H_u, y_u)\|^2.$$

Since $\eta > 0$, the loss $\ell(\omega; H_u, y_u)$ increases with each gradient ascent step, effectively reducing the contribution of the label $y_u$ to the active model $\omega$. Similarly, for the passive model $\theta$, we derive:

$$\begin{aligned}
\ell(\theta_{t+1}; x_u, y_u) &\approx \ell(\theta_t; x_u, y_u) + \nabla_\theta \ell(\theta_t; x_u, y_u)^\top (\theta_{t+1} - \theta_t) \\
&= \ell(\theta_t; x_u, y_u) + \eta \nabla_\theta \ell(\theta_t; x_u, y_u)^\top (\nabla_H \ell \nabla_\theta H) \\
&= \ell(\theta_t; x_u, y_u) + \eta \|\nabla_\theta \ell(\theta_t; x_u, y_u)\|^2,
\end{aligned}$$

where the first equation is due to the Eq. (6) of the main text and second equation is according to the chain rule. Thus, the contribution of the label $y_u$ is effectively removed from the passive model $\theta$.

2) *If the loss function $\ell$ is $\beta$-smooth*, we can further derive:

$$\begin{aligned}
\|\nabla_\omega \ell(\omega_T; H_u, y_u)\| &\leq \beta \|\omega_T - \omega_0\| \\
&= \|\sum_{t=0}^{T-1} \nabla_\omega \ell(\omega_t; H_u, y_u)\| \leq \beta\eta \sum_{t=0}^{T-1} \|\nabla_\omega \ell(\omega_t; H_u, y_u)\|,
\end{aligned} \tag{7}$$

where the second equation follows from Eq. (5) in the main text. **This result indicates that the convergence of gradient ascent depends on the learning rate** $\eta$. For instance, when the learning rate is small or includes a weight decay strategy(Patterson & Gibson, 2017), such as $\eta < \frac{1}{2\beta T}$, the gradient norm $\|\nabla_\omega \ell(\omega_T; H_u, y_u)\|$ tends to zero.

It is important to note that gradient ascent may impact the model utility on the remained data. To mitigate this, a small learning rate (smaller than $e^{-6}$ in Table 7 and 8) is adopted in this paper to minimize any decline in model utility for the remained data $D_r$. The experimental results presented in Section 5 validate this approach.

3) **The gradient ascent strategy aims to increase the model's loss corresponding to the unlearned label** $y_u$, thereby eliminating the contribution of the unlearned label $y_u$ to the model, as illustrated in 1).

## A.2   TABLE OF NOTATION

Table 5 summarises all the notations used in this paper.

## A.3   EXPERIMENTAL SETUP

**Datasets**   *MNIST*(Lecun et al., 1998) datasets contain images of handwritten digits.   MNIST dataset comprises 60,000 training examples and 10,000 test examples. Each example is represented as a single-channel image with dimensions of 28x28 pixels, categorised into one of 10 classes. *CIFAR10* (Krizhevsky et al., 2009) dataset comprises 60,000 images, each with dimensions of 32x32

| Notation | Meaning |
|---|---|
| $F_\omega, G_{\theta_k}$ | Active model and $k_{th}$ passive model |
| $K$ | The number of passive party |
| $\lambda$ | Mixed coefficient |
| $\eta$ | Learning rate |
| $N$ | Unlearning epochs |
| $\mathbf{x}_k$ | Private features own by $k_{th}$ passive party |
| $y$ | Private label own by active party |
| $y^u$ | The unlearn labels |
| $\{x_k^u\}$ | The unlearned feature for client $k$ corresponding to the $y^u$ |
| $x_k^p$ | The known features for client $k$ corresponding to the $y^u$ |
| $H_k$ | Forward embedding of passive party $k$ |
| $H_k'$ | Augmented forward embedding of passive party $k$ |
| $g_k'$ | Gradient on the embedding $H_k'$. |

Table 5: Table of Notations

pixels and three colour channels, distributed across 10 classes. This dataset includes 6,000 images per class and is partitioned into 50,000 training examples and 10,000 test examples. Within each class, there are 5000 training images and 1000 test images. Similarly, the *CIFAR100* (Krizhevsky et al., 2009) dataset shares the same image dimensions and structure as CIFAR10 but extends to 100 classes, with each class containing 600 images. Within each class, there are 500 training images and 100 test images. *ModelNet* (Wu et al., 2015) dataset is a widely-used 3D shape classification and shape retrieval benchmark, which currently contains 127,915 3D CAD models from 662 object categories. For the MNIST, CIFAR10, and CIFAR100 datasets, each image feature is divided among $K$ parties, where $K$ represents the number of passive parties. For the ModelNet dataset, we generate $K$ 2D multi-view images per 3D mesh model by placing two virtual cameras evenly distributed around the centroid. Each passive party is assigned one of the $K$ generated 2D multi-view images.

**Model Architecture**    Table 6 summarised our VFL framework settings.

| Model name | Model of Passive Party | Model of Active Party |
|---|---|---|
| Resnet18 | 20 Conv | 1 FC |
| Vgg16 | 13 Conv | 3 FC |

Table 6: Models in experiments. FC: Fully-connected layer. Conv: convolutional layer

**Implementation Details**    Table 7 and 8 summarise the hyper-parameters for our unlearning method.

| Hyper-parameters | Single-class | | | | | |
|---|---|---|---|---|---|---|
| | Resnet18-MNIST | Resnet18-CIFAR10 | Resnet18-CIFAR100 | Resnet18-ModelNet | Vgg16-CIFAR10 | Vgg16-CIFAR100 |
| Optimization Method | SGD | SGD | SGD | SGD | SGD | SGD |
| Unlearning Rate | 2e-7 | 2e-7 | 5e-7 | 5e-7 | 2e-7 | 5e-7 |
| Unlearning Epochs | 10 | 15 | 7 | 4 | 15 | 7 |
| Number of Data Samples | 40 | 40 | 30 | 30 | 40 | 30 |
| Batch Size | 32 | 32 | 32 | 32 | 32 | 32 |
| Weight Decay | 5e-4 | 5e-4 | 5e-4 | 5e-4 | 5e-4 | 5e-4 |
| Momentum | 0.9 | 0.9 | 0.9 | 0.9 | 0.9 | 0.9 |

Table 7: Hyper-parameters use for unlearning in our solution in Single-class unlearning.

Table 9 summarises the model name, datasets and unlearn classes involve in each unlearning scenarios.

| Hyper-parameters | Two-classes | | | | Multi-classes | |
|---|---|---|---|---|---|---|
| | Resnet18-CIFAR10 | Resnet18-CIFAR100 | Vgg16-CIFAR10 | Vgg16-Cifar100 | Resnet18-CIFAR100 | Vgg16-CIFAR100 |
| Optimization Method | SGD | SGD | SGD | SGD | SGD | SGD |
| Unlearning Rate | 1e-6 | 9e-7 | 1e-6 | 9e-7 | 9e-7 | 9e-7 |
| Unlearning Epochs | 15 | 10 | 15 | 5 | 15 | 5 |
| Number of Data Samples | 40 | 20 | 40 | 20 | 15 | 15 |
| Batch Size | 32 | 32 | 32 | 32 | 32 | 32 |
| Weight Decay | 5e-4 | 5e-4 | 5e-4 | 5e-4 | 5e-4 | 5e-4 |
| Momentum | 0.9 | 0.9 | 0.9 | 0.9 | 0.9 | 0.9 |

Table 8: Hyper-parameters use for unlearning in our solution in two-classes and multi-classes unlearning.

| Scenarios | Models | Datasets | Unlearn Classes |
|---|---|---|---|
| Single-class Unlearning | Resnet18 | MNIST, CIFAR10, CIFAR100, ModelNet | 0 |
| | Vgg16 | CIFAR10, CIFAR100 | 0 |
| Two-classes Unlearning | Resnet18 | CIFAR10, CIFAR100 | 0, 2 |
| | Vgg16 | CIFAR10, CIFAR100 | 0, 2 |
| Multi-classes Unlearning | Resnet18 | CIFAR100 | 0, 2, 5, 7 |
| | Vgg16 | CIFAR100 | 0, 2, 5, 7 |

Table 9: Models and datasets involve in each unlearning scenarios.

## A.4 ADDITIONAL EXPERIMENTS RESULTS

**Healthcare and NLP experiment.** We have incorporated one experiment using a healthcare dataset for classification task, specifically the Brain Tumor MRI dataset (Wang et al., 2024), which is commonly used in healthcare scenarios. The Brain Tumor MRI dataset consists of 7,023 human brain MRI images categorized into four classes: glioma, meningioma, no tumor, and pituitary.

Table 10 demonstrates that our method achieves strong unlearning effectiveness, with the accuracy on unlearned data ($D_u$) dropping from 95.67% to 2.43%. Furthermore, the accuracy on the remained data ($D_r$) outperforms other unlearning methods, except for retraining. For instance, the Amnesiac method results in an accuracy drop exceeding 20% while our method drops less than 10%. The decrease in the remained data accuracy for our method is attributed to the similarity of features among different labels. Removing one label can inadvertently impact the utility of other labels when using the gradient ascent method. In contrast, the retraining method performs well in maintaining the utility of other labels; however, it is significantly more time-consuming.

| Metrics | Accuracy (%) | | | | | | |
|---|---|---|---|---|---|---|---|
| | Baselines | Retrain | FT | Fisher | Amnesiac | BU | Ours |
| $\mathcal{D}_r$ | 97.92 | 98.81 ± 0.34 | 81.89 ± 0.82 | 30.26 ± 0.21 | 73.29 ± 0.09 | 45.30 ± 0.91 | **89.05 ± 0.61** |
| $\mathcal{D}_u$ | 95.67 | 0.00 ± 0.00 | 4.33 ± 0.49 | **0.00 ± 0.00** | **0.00 ± 0.00** | 3.67 ± 0.14 | 2.43 ± 0.04 |

Table 10: Single-label unlearning scenario with Brain MRI datasets on ResNet18 architecture. This experiments have one active party and two passive parties. The image features is split to half and each passive party own half of the image features. We unlearn label 0 (glioma) in this experiments.

Also, we have added experiments on Non-vision dataset (Yahoo Answers dataset (Fu et al., 2022a)) for the classification task. Yahoo Answers is a dataset designed for text classification tasks, comprising 10 classes (topics) such as "Society & Culture," "Science & Mathematics," "Health," "Education & Reference," among others. Each class contains 140,000 training samples and 6,000 testing samples. For simplicity, we utilized 5,000 training samples and 2,000 testing samples from each class.

Table 11 illustrates that our method performs effectively on both the accuracy of the remained data and the unlearned data. For instance, the unlearned data accuracy decreases from 41.63% to 5.14%, while the accuracy drop on the remained data is less than 3%.

**More passive parties.** In addition, we conducted experiments with one active party and eight passive parties on the CIFAR-10 dataset using the ResNet-18 architecture. The image features were split into eight parts, with each passive party owning one-eighth of the image features. Table 13 and Figure 9 demonstrates that the proposed method continues to perform well in terms of both unlearn-

| Metrics | Accuracy (%) | | |
|---|---|---|---|
| | Baseline | Retrain | Ours |
| $\mathcal{D}_r$ | 62.92 | $63.14 \pm 0.45$ | $60.72 \pm 0.98$ |
| $\mathcal{D}_u$ | 41.63 | $0.00 \pm 0.00$ | $5.14 \pm 1.04$ |

Table 11: Single-label unlearning scenario on Yahoo Answer datasets with MixText architecture ((Chen et al., 2020), transformer-based models). This experiments have one active party and two passive parties. Each sample (a single paragraph of text) is divided into two paragraphs, with each passive party holding one of them. We unlearn label 6 (Business & Finance) in this experiments.

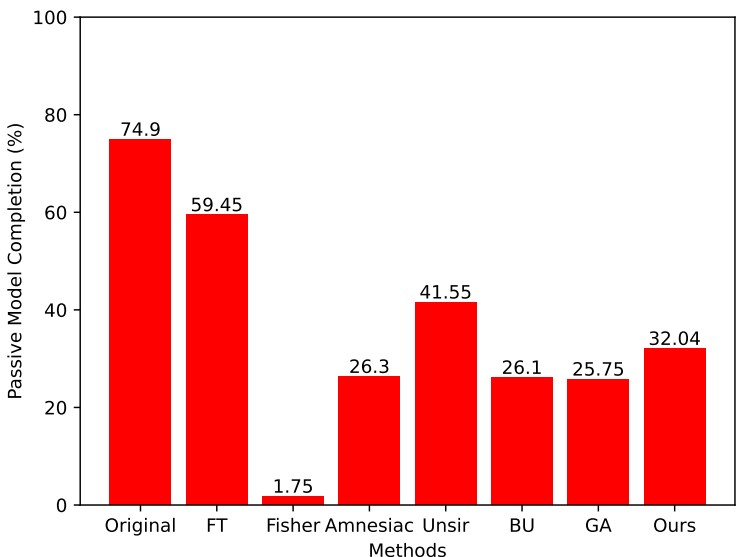

Figure 8: PMC resnet18 cifar10 single class

ing effectiveness and the utility of the remained data. For instance, the accuracy on the unlearned data drops to 0.17%, while the accuracy on the remained data decreases by less than 3%.

**PMC attack.** Moreover, Figure 8 shows the PMC attack (one strongest label privacy attack in (Fu et al., 2022b)) before and after unlearning methods. It demonstrates that our methods achieve beyond 40% drops for the model accuracy on $D_u$.

**Efficiency for more passive parties.** The manifold mixup step is executed by each passive party, rather than the active party (see Figure 3 and Algorithm 1 of the main text). As a result, the unlearning time increases linearly with the number of passive parties. The unlearning times of different methods are compared for varying numbers of passive parties in the table below, demonstrating that our method remains the most efficient compared to the alternatives.

**Ablation study for $\lambda$.** For each dataset used in this paper, we augment the embeddings with two coefficients, i.e., $\lambda = 0.25$ and $\lambda = 0.5$. Additionally, we evaluate the impact of different $\lambda$ values in Table 12. The results indicate that variations in $\lambda$ have a minimal impact on the unlearning effectiveness.

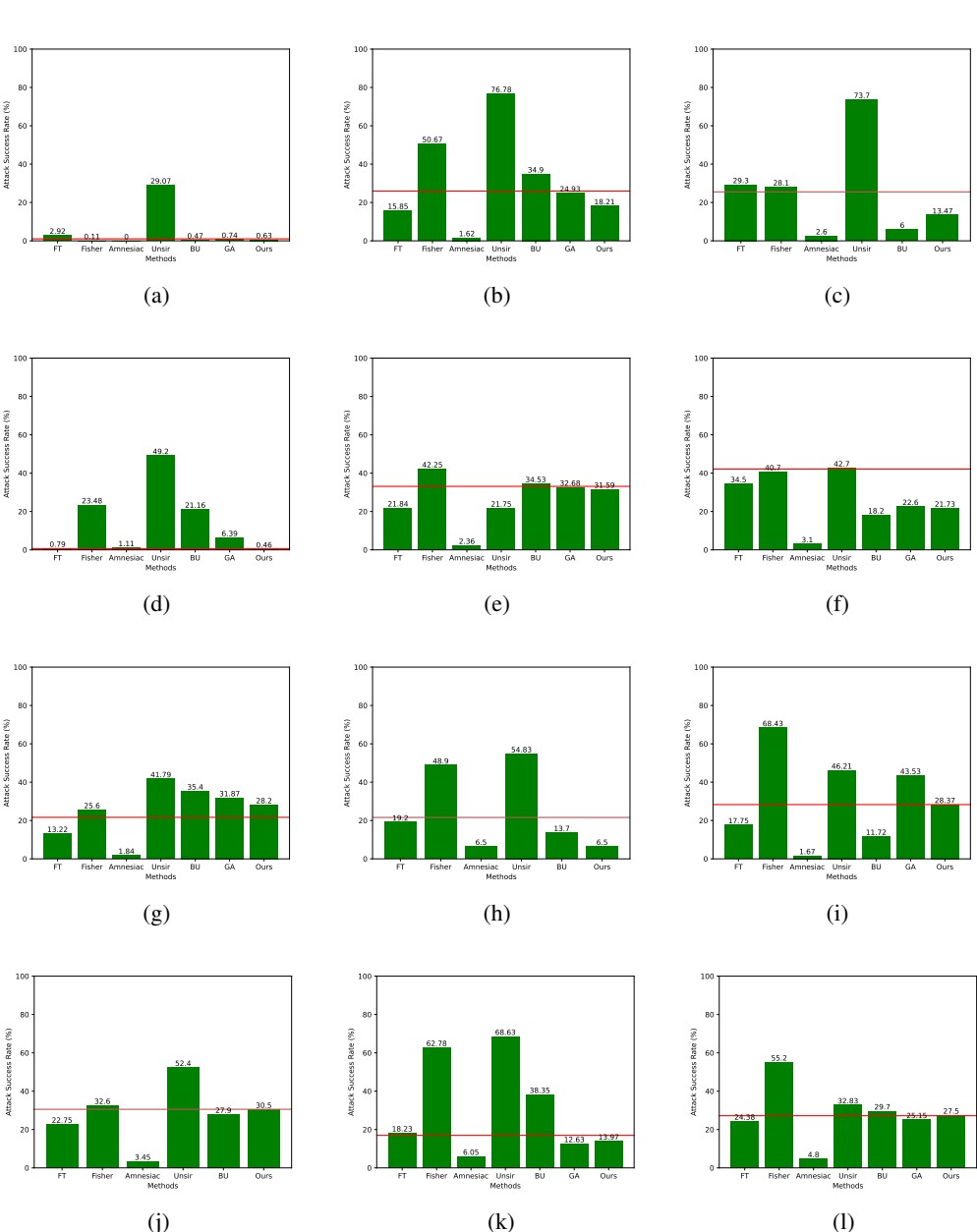

Figure 9: The following sub-figures show the MIA attack success rate on (a) Single-class Resnet18 Mnist, (b) Single-class Resnet18 Cifar10, (c) Single-class Resnet18 Cifar100, (d) Single-class Resnet18 ModelNet, (e) Single-class Vgg16 Cifar10, (f) Single-class Vgg16 Cifar100, (g) Two-classes Resnet18 Cifar10, (h) Two-classes Resnet18 Cifar100, (i) Two-classes Vgg16 Cifar10, (j) Two-classes Vgg16 Cifar100, (k) Multi-classes Resnet18 Cifar100, (l) Multi-classes Vgg16 Cifar100. The red line in graphs represent the ASR of retrained model.

| $\lambda$ Rate | Metrics | Accuracy (%) |
|---|---|---|
| [0.2, 0.5] | $\mathcal{D}_r$ | $88.69 \pm 0.19$ |
| | $\mathcal{D}_u$ | $1.77 \pm 0.57$ |
| [0.25, 0.5] | $\mathcal{D}_r$ | $89.11 \pm 0.14$ |
| | $\mathcal{D}_u$ | $0.00 \pm 0.00$ |
| [0.33, 0.5] | $\mathcal{D}_r$ | $88.78 \pm 0.09$ |
| | $\mathcal{D}_u$ | $2.10 \pm 0.42$ |

Table 12: Different lambda rate on single-label unlearning scenarios on CIFAR10 dataset with ResNet18 architecture. We unlearn label 0 in this experiment.

| Metrics | Accuracy (%) | | | | | | |
|---|---|---|---|---|---|---|---|
| | Baseline | Retrain | Fisher | Amnesiac | Unsir | BU | Ours |
| $\mathcal{D}_r$ | 84.16 | $84.98 \pm 0.11$ | $18.01 \pm 0.38$ | $77.28 \pm 0.93$ | $67.95 \pm 0.86$ | $70.99 \pm 0.70$ | $82.72 \pm 0.99$ |
| $\mathcal{D}_u$ | 87.9 | $0.00 \pm 0.00$ | $0.00 \pm 0.00$ | $0.00 \pm 0.00$ | $0.00 \pm 0.00$ | $0.50 \pm 0.07$ | $0.17 \pm 0.03$ |

Table 13: Single-label unlearning scenario on CIFAR10 dataset with Resnet18 architecture on 8 passive parties. The image features is equally split into 8 parts and each passive party own one eight of the image features. We unlearn label 0 in this experiment.

| # of Passive Parties | Runtime (s) | | | | | | |
|---|---|---|---|---|---|---|---|
| | Retrain | FT | Fisher | Amnesiac | Unsir | BU | Ours |
| 1 | $3008.69 \pm 1.69$ | $134.05 \pm 0.01$ | $197.35 \pm 0.51$ | $95.29 \pm 0.47$ | $48.89 \pm 0.12$ | $43.59 \pm 0.14$ | $\mathbf{1.52 \pm 0.04}$ |
| 2 | $3725.23 \pm 8.17$ | $167.11 \pm 0.38$ | $254.51 \pm 5.98$ | $122.79 \pm 0.22$ | $55.52 \pm 0.45$ | $49.48 \pm 0.59$ | $\mathbf{2.94 \pm 0.35}$ |
| 4 | $5647.67 \pm 2.42$ | $361.34 \pm 2.47$ | $401.33 \pm 3.79$ | $203.68 \pm 1.32$ | $78.39 \pm 0.41$ | $82.71 \pm 3.06$ | $\mathbf{3.48 \pm 0.02}$ |
| 8 | $9699.87 \pm 10.37$ | $539.27 \pm 4.02$ | $847.71 \pm 1.89$ | $201.55 \pm 3.53$ | $138.34 \pm 0.82$ | $159.09 \pm 0.99$ | $\mathbf{7.04 \pm 0.44}$ |

Table 14: Comparison of runtime between 1,2,4, and 8 passive parties.

