# OpenReview forum: "A few-shot Label Unlearning in Vertical Federated Learning"
_ICLR.cc/2025/Conference — Submitted to ICLR 2025_

### Official Review · Reviewer_ame3 · 2024-10-23

**Soundness:** 3
**Presentation:** 3
**Contribution:** 3
**Rating:** 8
**Confidence:** 4

**Summary:**

This paper explores the unlearning process in vertical federated learning (VFL), a topic that has been relatively underexplored in prior research. The authors highlight the risk of label leakage during the unlearning phase and propose a solution to address this issue. Their method utilizes a small set of labeled data, combined with feature mixup and gradient ascent, to mitigate the leakage of label information from VFL systems.

**Strengths:**

1. The proposed method is simple and intuitive, approaching the unlearning process from a few-shot perspective and utilizing feature mixup to handle limited sample sizes.
2. The method demonstrates broad applicability.
3. Extensive experimental results indicate that the method not only performs consistently across various datasets and models with different numbers of clients but can also be effectively integrated with differential privacy and gradient compression methods. Furthermore, the proposed method exhibits the lowest unlearning cost compared to its competitors.

**Weaknesses:**

1. The label leakage described in Sec. 3.2 would only occur if both active and passive clients share the same label space. However, in most VFL scenarios, clients have distinct label spaces. Therefore, this assumption of label leakage is questionable.
2. In the introduction, the statement that "... the passive party holds the features, ... the active party possesses the labels." is inaccurate. The active party also holds its own features.
3. Some notations are unclear. For instance, in line 152, "{x^u_k}" is used, but the variable "u" is not defined. I assume it refers to samples for unlearning.
4. The figures are difficult to interpret, particularly Fig. 1 and Fig. 6.
5. The mix rate lambda is crucial to the method. However, the authors neither provide a thorough analysis of this hyperparameter nor clarify whether lambda remains consistent across different clients.

**Questions:**

Please check the weakness part for details.

---

> ### Author Response · Authors · 2024-11-22
>
> **1)The label leakage described in Sec. 3.2 would only occur if both active and passive clients share the same label space. However, in most VFL scenarios, clients have distinct label spaces. Therefore, this assumption of label leakage is questionable.**
>
> In VFL, the participating parties hold datasets that share a common set of sample IDs but differ in feature spaces. Let's take a real-world bank and insurance company collaboration as an example of VFL.
> Bank (Active Party) holds customer labels (e.g., whether the customer defaults on a loan: 1 for default, 0 otherwise) and features like transaction history or credit score. Insurance Company (Passive Party)
> holds customer features like their insurance claims history and premium payment records, but does not have access to labels.
> The goal is to collaboratively train a model to predict loan defaults while preserving the privacy of both parties. **Therefore, the two parties have distinct feature spaces but share the same label space (e.g., lending decision) [6,7].**
>
> Moreover, when the bank aims to unlearn the label, it seeks to ensure that the label (e.g., lending decision) is not disclosed to the insurance company, as this information is both critical and sensitive.
>
> **2)In the introduction, the statement that "... the passive party holds the features, ... the active party possesses the labels." is inaccurate. The active party also holds its own features.**
>
> We apologize for the inaccurate clarification. According to the definition of VFL [6, 7], the active party may also possess features. We will revise the corresponding sentences in our paper to reflect this definition accurately.
>
> Furthermore, **if the active party has features, the training process for these features is identical to that for the passive parties**. Specifically, the active party first computes the feature embeddings and then concatenates these embeddings with the feature embeddings transferred by the passive parties. Consequently, the experimental setup in this paper—where the active party owns the label and two passive parties own the features—is equivalent to a scenario where one passive party owns the features while the active party owns both features and the label.
>
> **3)Some notations are unclear. For instance, in line 152, "${x^{u}_{k}}$" is used, but the variable "u" is not defined. I assume it refers to samples for unlearning.**
>
> We apologize for the unclear definition. Here, ${x^{u}_{k}}$ represents the unlearned features for client $k$. In response, we will include a notation table in the future version of the paper, as outlined in Reply 1 to Reviewer 3.
>
>
> **4)The figures are difficult to interpret, particularly Fig. 1 and Fig. 6.**
>
> We acknowledge the lack of clarity in our previous explanation.
>
> Figure 1 illustrates the risk of label leakage during the unlearning process in Vertical Federated Learning (VFL). In this process, the active party aims to unlearn the label information stored in the passive model owned by the passive party by transferring certain information, such as gradients. However, this exchange may inadvertently reveal sensitive information, allowing the passive party to infer the label of the active party.
>
> Figure 6 compares the unlearning performance across different sizes of labeled data $D_p$. The results indicate that when using a limited amount of data ($|D_p| = 40$), directly applying gradient ascent (GA-s) does not achieve satisfactory unlearning effectiveness, as the accuracy on the unlearned data remains at 40.48\%. In contrast, our method, which incorporates manifold mixup, demonstrates significantly better unlearning effectiveness. For instance, with only 40 labeled data points, our approach reduces the unlearned accuracy to 0\%.
>
>
> [6] Liu Y, Kang Y, Zou T, et al. Vertical federated learning: Concepts, advances, and challenges[J]. IEEE Transactions on Knowledge and Data Engineering, 2024.
>
> [7] Yang Q, Liu Y, Chen T, et al. Federated machine learning: Concept and applications[J]. ACM Transactions on Intelligent Systems and Technology (TIST), 2019, 10(2): 1-19.

---

> ### Author Response · Authors · 2024-11-22
>
> **5)The mix rate lambda is crucial to the method. However, the authors neither provide a thorough analysis of this hyperparameter nor clarify whether lambda remains consistent across different clients.**
>
> For each dataset used in this paper, we augment the embeddings with two coefficients, i.e., $\lambda = 0.25$ and $\lambda = 0.5$. Additionally, we evaluate the impact of different $\lambda$ values in Table 5. The results indicate that variations in $\lambda$ have a minimal impact on the unlearning effectiveness.
>
> | $\lambda$ Rate| Metrics | Accuracy (%) |
> |:-------:|:------------:|:---------------:|
> |[0.2,0.5] |$\mathcal{D}_{r}$|88.69&pm;0.19|
> ||$\mathcal{D}_{u}$|1.77&pm;0.57|
> |[0.25, 0.5]|$\mathcal{D}_{r}$|89.11&pm;0.14|
> ||$\mathcal{D}_{u}$|0.00&pm;0.00|
> |[0.33, 0.5]|$\mathcal{D}_{r}$ |88.78&pm;0.09|
> ||$\mathcal{D}_{u}$ |2.10&pm;0.42|
>
> Table 5 : Different lambda rate on single-label unlearning scenarios on CIFAR10 dataset with ResNet18 architecture. We unlearn label 0 in this experiment.

---

> > ### Comment · Reviewer_ame3 · 2024-11-25
> >
> > Thank you for your effort. The rebuttal adequately addressed my concerns. After carefully reviewing the questions and responses raised by other reviewers, I find this work to be solid and intriguing. Therefore, I would like to raise my score to support its acceptance.

---

> ### Author Response · Authors · 2024-11-25
> **Thank You for Rating Upgrade**
>
> Thank you for your thoughtful feedback. We are delighted to hear that our rebuttal has addressed most of your concerns. We deeply appreciate your decision to revise and elevate the rating to **8 (Accept)**. Your acknowledgment of our revisions is incredibly meaningful to us, and we remain fully committed to addressing any additional concerns you may have.
>
> Should there be any further points for discussion, we are more than willing to engage in dialogue to ensure that our paper meets the highest standards of quality and rigor.
>
> Once again, thank you for your invaluable insights and for your thoughtful consideration in upgrading our score.

---

### Official Review · Reviewer_mNtD · 2024-11-03

**Soundness:** 3
**Presentation:** 2
**Contribution:** 3
**Rating:** 5
**Confidence:** 4

**Summary:**

This paper proposes a few-shot label unlearning method specifically designed for Vertical Federated Learning (VFL) to mitigate privacy leakage risks associated with traditional unlearning approaches. In VFL, multiple organizations collaborate on model training by sharing some sample IDs while differing in feature sets, making it particularly suitable for sensitive fields such as banking and healthcare. This method leverages a small amount of private data and manifold mixup to augment embeddings, employing gradient ascent to remove labels from the active party’s model and transferring inverse gradients to the passive party to ensure independent label removal from both models. Experimental results demonstrate that this approach achieves rapid and effective label unlearning across various datasets, significantly reducing privacy leakage risks.

**Strengths:**

1. **Introducing the first method for label unlearning in Vertical Federated Learning (VFL)**: This paper pioneers a novel approach specifically tailored for the removal of label information in VFL settings, addressing a critical gap in existing federated learning literature. Unlike prior work focused on feature-based unlearning, this method uniquely targets the secure and efficient unlearning of labels across collaborative models, enhancing data protection compliance in sensitive domains.

2. **Detailing potential privacy leaks in traditional unlearning methods**: The paper systematically examines the vulnerabilities of conventional unlearning techniques when applied to VFL, especially the risk of sensitive label information being inadvertently exposed to other parties during the unlearning process. By identifying specific points of leakage, this study highlights the importance of secure information flow between the active and passive parties in VFL.

3. **Proposing an effective few-shot unlearning approach that uses limited data and enhances unlearning with manifold mixup**: To minimize data exposure, this approach relies on a small subset of private data, significantly reducing privacy risks. Through manifold mixup, the method augments data embeddings, allowing for robust and efficient unlearning with minimal information. This few-shot strategy achieves a balance between privacy preservation and computational efficiency, making it practical for real-world applications.

4. **Demonstrating superior unlearning performance on benchmark datasets like MNIST, CIFAR-10, CIFAR-100, and ModelNet**: Extensive experiments validate the method's effectiveness and speed in removing target labels, outperforming traditional unlearning techniques on widely used datasets. The results illustrate the approach’s scalability and adaptability across diverse data types, affirming its potential for high-performance unlearning in federated settings.

**Weaknesses:**

1. The paper lacks a preliminary section to help readers understand the meaning of each mathematical symbol. Some symbols are not explained, making it difficult for me to follow the methodology. Please clarify the definition of each variable in the paper.

2. While the paper’s motivation is clear and innovative, it lacks explanations regarding specific methods, such as why gradient ascent and inverse gradient can remove label information. Additionally, I am uncertain whether the proposed method guarantees convergence. Please provide an analysis or discussion on the forgetting functionality and convergence of this method.
(1). provide a theoretical justification for why gradient ascent and inverse gradients are effective for label removal
(2). Include a convergence analysis or discussion of the conditions under which convergence is guaranteed
(3). Offer more detailed explanations of the forgetting mechanism and how it ensures complete removal of label information

3. The experimental setup lacks sufficient detail, including which categories were selected for forgetting in each dataset, the number of clients, and how data attributes were assigned. Please add a more detailed description of the experimental setup.
(1). A table or list of the exact categories chosen for forgetting in each dataset
(2). The number of clients/parties involved in each experiment
(3). A description of how data attributes were distributed among the parties This level of detail would allow for better reproducibility and understanding of the experimental conditions.

**Questions:**

See weakness

---

> ### Author Response · Authors · 2024-11-22
>
> **1)The paper lacks a preliminary section to help readers understand the meaning of each mathematical symbol**
>
> Thank you for your suggestions. We plan to include the following table in a future version:
> | Notations | Meanings |
> |:-------:|:------------:|
> |$F_{\omega}, G_{\theta_k}$ | Active model and $k_{th}$ passive model |
> |$K$| The number of passive party  |
> |$\lambda$|Mixed coefficient|
> |$\eta$| Learning rate|
> |$N$ | Unlearning epochs|
> |$\mathbf{x}_k$| Private features own by $k_{th}$ passive party|
> |$y$   |  Private label own by active party|
> |$y^u$ |  The unlearn labels|
> |$x_k^u$ | The unlearned feature for client $k$ corresponding to the $y^u$|
> |$x_k^p$ | The  known features for client $k$ corresponding to the $y^u$|
> |$H_k$  |  Forward embedding of passive party $k$|
> |$H_k'$ |  Augmented forward embedding of passive party $k$|
> |$g_k'$ | Gradient on the embedding $H_k'$ |
> Table 4 : Table of Notations
>
> **2) (1). provide a theoretical justification for why gradient ascent and inverse gradients are effective for label removal (2). Include a convergence analysis or discussion of the conditions under which convergence is guaranteed (3). Offer more detailed explanations of the forgetting mechanism and how it ensures complete removal of label information**
>
> We sincerely appreciate your constructive feedback.
> Below, we address the questions systematically. Consider a scenario where the active party seeks to unlearn the label $y'$ with the corresponding feature $x'$ and embedding $H' = G_\theta(x')$. The gradient ascent approach aims to remove the label information $y'$ from both the active model $\theta$ and the passive model $\omega$.
>
> 1) **Unlearning effectiveness for Gradient Ascent (GA).** Using the first-order Taylor expansion of $ \ell(\omega; H', y') $ around the initial parameter $ \omega_t $, we obtain:
>
> $\ell(\omega_{t+1}; H', y') \approx \ell(\omega_t; H', y') + \nabla_\omega \ell(\omega_t; H', y')^\top (\omega_{t+1} - \omega_t)$
>
> Substituting the gradient ascent update $ \omega_{t+1} = \omega_t + \eta \nabla_\omega \ell(\omega_t; H', y') $ (as defined in Eq. (5) of the main text), this becomes:
>
> $\ell(\omega_{t+1}; H', y') \approx \ell(\omega_t; H', y') + \eta \|\nabla_\omega \ell(\omega_t; H', y')\|^2$
>
> Since $ \eta > 0 $, the loss $ \ell(\omega; H', y') $ increases with each gradient ascent step, effectively reducing the contribution of the label $y'$ to the active model $\omega$. Similarly, for the passive model $\theta$, we derive:
> \begin{equation*}
> \begin{split}
> \ell(\theta_{t+1}; x', y') & \approx \ell(\theta_t; x', y') + \nabla_\theta \ell(\theta_t; x', y')^\top (\theta_{t+1} - \theta_t)\\
> \end{split}
> \end{equation*}
> \begin{equation*}
> \begin{split}
>  & = \ell(\theta_t; x', y') + \eta \nabla_\theta \ell(\theta_t; x', y')^\top (\nabla_H \ell \nabla_\theta H) \\
> \end{split}
> \end{equation*}
> \begin{equation*}
> \begin{split}
>  & = \ell(\theta_t; x', y') + \eta \|\nabla_\theta \ell(\theta_t; x', y')\|^2,
> \end{split}
> \end{equation*}
> where the first equation is due to the Eq. (6) of the main text and second equation is according to the chain rule. Thus, the contribution of the label $y'$ is effectively removed from the passive model $\theta$.
>
> 2)*If the loss function $\ell$ is $\beta$-smooth*, we can further derive:
> \begin{equation}
> \begin{split}
>        & \|\nabla_\omega \ell(\omega_T; H', y')\| \leq \beta \| \omega_T - \omega_0 \|  \\
>         & =\| \sum_{t=0}^{T-1} \nabla_\omega \ell(\omega_t; H', y')\|
>         \leq \beta \eta \sum_{t=0}^{T-1} \|\nabla_\omega \ell(\omega_t; H', y')\|,
>         \end{split}
> \end{equation}
> where the second equation follows from Eq. (5) in the main text. \textbf{This result indicates that the convergence of gradient ascent depends on the learning rate $\eta$}. For instance, when the learning rate is small or includes a weight decay strategy [5], such as $\eta < \frac{1}{2\beta T}$, the gradient norm $\|\nabla_\omega \ell(\omega_T; H', y')\|$ tends to zero.
>
> It is important to note that gradient ascent may impact the model utility on the remained data. To mitigate this, a small learning rate (smaller than $e^{-6}$ in Appendix A) is adopted in this paper to minimize any decline in model utility for the remained data $D_r$. The experimental results presented in Table 4 validate this approach.
>
> [5] Patterson, Josh; Gibson, Adam (2017). "Understanding Learning Rates". Deep Learning : A Practitioner's Approach. O'Reilly. pp. 258–263
>
> 3) **The gradient ascent strategy aims to increase the model's loss corresponding to the unlearned label $y'$**, thereby eliminating the contribution of the unlearned label $y'$ to the model, as illustrated in 1).
>
> We will also add the above discussion in the future version.

---

> ### Author Response · Authors · 2024-11-22
>
> **The experimental setup lacks sufficient detail, including which categories were selected for forgetting in each dataset, the number of clients, and how data attributes were assigned. Please add a more detailed description of the experimental setup. (1). A table or list of the exact categories chosen for forgetting in each dataset (2). The number of clients/parties involved in each experiment (3). A description of how data attributes were distributed among the parties This level of detail would allow for better reproducibility and understanding of the experimental conditions.**
>
> We sincerely appreciate your constructive suggestions. In response, we provide the following explanation and will include it in Appendix A:
>
> 1) The classes selected for unlearning remain consistent across all datasets. Specifically: a) In single-label unlearning, we unlearn label 0. b) In two-label unlearning, we unlearn labels 0 and 2. c) In multi-label unlearning, we unlearn labels 0, 2, 5, and 7.
>
> 2) Unless otherwise specified, all experiments in this paper involve two passive parties and one active party. In Table 4 of the main text, we consider setups with one active party and one, two, or four passive parties.
>
>
> 3) In our experiments, the passive party owns the feature $x$, while the active party owns the label $y$. For the MNIST, CIFAR10, and CIFAR100 datasets, each image feature is divided among $K$ parties, where $K$ represents the number of passive parties. For the ModelNet dataset, we generate $K$ 2D multi-view images per 3D mesh model by placing two virtual cameras evenly distributed around the centroid. Each passive party is assigned one of the  $K$ generated 2D multi-view images.

---

> ### Author Response · Authors · 2024-12-02
> **Follow up Rebuttal**
>
> As today marks the final opportunity for reviewers to share any additional feedback with the authors, I wanted to kindly follow up to check if you have any further questions or require additional clarifications regarding our rebuttal. We would be delighted to address any remaining concerns promptly within the timeframe available (by tomorrow).
>
> We sincerely hope our responses have effectively addressed your valuable feedback. If they meet your expectations, we would greatly appreciate it if you could reflect this in your rating.
>
> Once again, we deeply appreciate the time, effort, and expertise you have dedicated to reviewing our work. Your constructive comments have been instrumental in enhancing the quality of our submission, and we are truly grateful for your guidance throughout this process.
>
> Thank you for your continued support.

---

### Official Review · Reviewer_45aD · 2024-11-04

**Soundness:** 4
**Presentation:** 4
**Contribution:** 4
**Rating:** 6
**Confidence:** 4

**Summary:**

This paper focuses on the unlearning problem in vertical federated learning (VFL). The authors leverage a limited amount of data for unlearning to mitigate the label leakage risk of existing unlearning methods. In particular, the authors propose to mixup feature embeddings during the learning process. Then during the unlearning process, the mixup scheme can be utilized to augment the limited data. The unlearning is finally achieved by gradient ascent on the augmented data. Experiments conducted on diverse datasets validate the performance of the proposed framework.

**Strengths:**

* The discussed topic, unlearning in VFL, is important.
* The writing is clear.
* The proposed mixup augmentation is simple yet effective.
* The empirical results demonstrate the effectiveness of the proposed framework. The model after unlearning demonstrate high accuracy and low attack success rate.
* The ablation is extensive and helpful.

**Weaknesses:**

As shown in Equation (3), the feature size of the active party increases quadratically as the number of clients increases. Therefore, the scalability of the framework is relatively limited.

The effeciency of the proposed method is still doubted.
The authors did not correct their submission.
I will maintain my score.

**Questions:**

As shown in Figure 4, the ASR of Amnesiac is abnormally high (up to 96.1) on CIFAR-100 with Resnet 18 architecture while ASR of Amnesiac on other datasets or architectures is only up to 3.1. Could the authors provide more explanations?

---

> ### Author Response · Authors · 2024-11-22
>
> **1) As shown in Equation (3), the feature size of the active party increases quadratically as the number of clients increases. Therefore, the scalability of the framework is relatively limited.**
>
> Thank you for your suggestion. VFL is primarily applied in collaborations between different companies. As a result, the number of participating parties is typically limited, distinguishing it from Horizontal Federated Learning (HFL), which often involves thousands of clients [5].
>
> [5] Liu Y, Kang Y, Zou T, et al. Vertical federated learning: Concepts, advances, and challenges[J]. IEEE Transactions on Knowledge and Data Engineering, 2024.
>
> In addition, we conducted experiments with one active party and eight passive parties on the CIFAR-10 dataset using the ResNet-18 architecture. The image features were split into eight parts, with each passive party owning one-eighth of the image features. Table 3 below demonstrates that the proposed method continues to perform well in terms of both unlearning effectiveness and the utility of the remained data. For instance, the accuracy on the unlearned data drops to 0.17\%, while the accuracy on the remained data decreases by less than 3\%.
>
> | Metrics | Accuracy (%) |                 |                    |                    |                    |                    |                 |                 |
> |:-------:|:------------:|:---------------:|:------------------:|:------------------:|:------------------:|:------------------:|:---------------:|:---------------:|
> | Metrics |   Baselines  |     Retrain     |      Fisher       |      Amnesiac      |        Unsir       |        BU       |       Ours      |
> |    $\mathcal{D}_{r}$   |     84.16    | 84.98 &pm; 0.11 |    18.01 &pm; 0.38  |   77.28.29 &pm; 0.93  |   67.95 &pm; 0.86  | 70.99 &pm; 0.70 | **82.72 &pm; 0.99** |
> |     $\mathcal{D}_{u}$    |     87.9    |  0.00 &pm; 0.00 |  **0.00 &pm; 0.00** | **0.00 &pm; 0.00** | **0.00 &pm; 0.00** |  0.50 &pm; 0.07 |  0.17 &pm; 0.03 |
>
> Table 3 : Single-label unlearning scenario on CIFAR10 dataset with Resnet18 architecture on 8 passive parties. The image features is equally split into 8 parts and each passive party own one eight of the image features. We unlearn label 0 in this experiment.
>
> $\newline$
> **2) As shown in Figure 4, the ASR of Amnesiac is abnormally high (up to 96.1) on CIFAR-100 with Resnet 18 architecture while ASR of Amnesiac on other datasets or architectures is only up to 3.1. Could the authors provide more explanations?**
>
> We apologize for the typo. The correct Attack Success Rate (ASR) value of Amnesiac for ResNet18 on the CIFAR100 dataset is 2.6\%. Specifically, we remove the gradients using a batch size of 64 for Amnesiac unlearning.

---

> ### Comment · Reviewer_45aD · 2024-11-27
> **Thanks for response**
>
> The effeciency of the proposed method is still doubted. The authors did not correct their submission. Therefore, I will maintain my score.

---

> ### Author Response · Authors · 2024-11-28
>
> Thank you, we appreciate the opportunity to clarify further.
>
> The manifold mixup step is executed by each passive party, rather than the active party (see Figure 3 and Algorithm 1 of the main text). As a result, **the unlearning time increases linearly with the number of passive parties.** The unlearning times of different methods are compared for varying numbers of passive parties in the table below, demonstrating that our method remains the most efficient compared to the alternatives.
>
> **Runtime comparison**
> | Number of Passive Parties |   Runtime (s)  |              |              |              |              |              |            |
> |:-------------------------:|:--------------:|:------------:|:------------:|:------------:|:------------:|:------------:|:----------:|
> |                           |     Retrain    |      FT      |    Fisher    |   Amnesiac   |     Unsir    |      BU      |    Ours    |
> |             1             |  3008.69 &pm; 1.69 | 134.05 &pm; 0.01 | 197.35 &pm; 0.51 |  95.29 &pm; 0.47 |  48.89 &pm; 0.12 |  43.59 &pm; 0.14 | **1.52 &pm; 0.04** |
> |             2             |  3725.23 &pm; 8.17 | 167.11 &pm; 0.38 | 254.51 &pm; 5.98 | 122.79 &pm; 0.22 |  55.52 &pm; 0.45  |  49.48 &pm; 0.59 | **2.94 &pm; 0.35** |
> |             4             |  5647.67 &pm; 2.42 | 361.34 &pm; 2.47 | 401.33 &pm; 3.79 | 203.68 &pm; 1.32 |  78.39 &pm; 0.41 |  82.71 &pm; 3.06 | **3.48 &pm; 0.02** |
> |             8             | 9699.87 &pm; 10.37 | 539.27 &pm; 4.02 | 847.71 &pm; 1.89 | 201.55 &pm; 3.53 | 138.34 &pm; 0.82 | 159.09 &pm; 0.99 | **7.04 &pm; 0.44** |
>
> Table 6 : Comparison of unlearning runtime between 1,2,4, and 8 passive parties with ResNet18 on CIFAR10.
>
> Furthermore, as defined in Equation (3), the manifold mixup operation is applied to the data $D_p$, **leading to a quadratic increase in unlearning time relative to the size of $D_p$.** In our experiments, the size of $D_p$ is fixed at a small value of 40. Tables 1 and 2, along with Figure 4 of the main text, illustrate that our method $|D_p| =40$ consistently excels in unlearning effectiveness, model utility, and unlearning efficiency.

---

> ### Author Response · Authors · 2024-11-28
>
> We have revised and uploaded a new version (PDF), which incorporates the changes based on the official comments we provided above.
>
> Thank you.

---

> ### Author Response · Authors · 2024-12-02
> **Follow up rebuttal  2**
>
> As today marks the final opportunity for reviewers to share any additional feedback with the authors, I wanted to kindly follow up to check if you have any further questions or require additional clarifications regarding our new rebuttal above on **efficiency** and **PDF update**. We would be delighted to address any remaining concerns promptly within the timeframe available (by tomorrow).
>
> We hope our new updated responses have effectively addressed your valuable feedback. If they meet your expectations, we would greatly appreciate it if you could reflect this in your rating.
>
> Once again, we deeply appreciate the time, effort, and expertise you have dedicated to reviewing our work. Your constructive comments have been instrumental in enhancing the quality of our submission, and we are truly grateful for your guidance throughout this process.
>
> Thank you for your continued support.

---

### Official Review · Reviewer_7Ypp · 2024-11-04

**Soundness:** 3
**Presentation:** 3
**Contribution:** 3
**Rating:** 5
**Confidence:** 2

**Summary:**

This paper aims to address the label unlearning problem in vertical federated learning (VFL). It demonstrates the lable leakage issue with applying traditional techniques in lable unlearning in VFL and proposes a few shot unlearning method to address mitigate this issue. Extensive experiments on 4 datasets and 2 model architectures show that the proposed method is effective and effcient while preserving good utility.

**Strengths:**

- The paper is well structured and generally easy to follow
- The proposed method is intuitive and seems to be technically sound
- Extensive experiments on 4 datasets and 2 model architectures demonstrate the effectiveness of the proposed method

**Weaknesses:**

- As mentioned in the introduction, VFL is mostly used in finance, healthcare, or ecommerce scenarios, but this work mostly validated the proposed method on simple vision datasets.
- The discussion and presentation of experiment results can be improved. e.g., the paper discusses utility in Sec 5.2.1 but also mentioning effectiveness in Figure 4 in Sec 5.2.2. It is not intuitive to compare different methods on these two aspects easily.
- Figure 1 is not easy to understand and interpret the label leakage risk.
- VFU seems to be used without definition

**Questions:**

- Can the proposed method be applied to non-vision datasets or other vision tasks other than classification?

---

> ### Author Response · Authors · 2024-11-22
>
> **1)As mentioned in the introduction, VFL is mostly used in finance, healthcare, or ecommerce scenarios, but this work mostly validated the proposed method on simple vision datasets.**
>
> Thank you for your suggestion. We have incorporated one experiment using a healthcare dataset for classification task, specifically the Brain Tumor MRI dataset [1], which is commonly used in healthcare scenarios. The Brain Tumor MRI dataset consists of 7,023 human brain MRI images categorized into four classes: glioma, meningioma, no tumor, and pituitary.
>
> Table 1 below demonstrates that our method achieves strong unlearning effectiveness, with the accuracy on unlearned data ($D_u$) dropping from 95.67\% to 2.43\%. Furthermore, the accuracy on the remained data ($D_r$) outperforms other unlearning methods, except for retraining. For instance, the Amnesiac method results in an accuracy drop exceeding 20\% while our method drops less than 10\%. The decrease in the remained data accuracy for our method is attributed to the similarity of features among different labels. Removing one label can inadvertently impact the utility of other labels when using the gradient ascent method. In contrast, the retraining method performs well in maintaining the utility of other labels; however, it is significantly more time-consuming.
>
> [1] Wang, Z., Gao, X., Wang, C., Cheng, P., Chen, J., Zichen WangZhejiang University, H., Xiangshan GaoZhejiang University, H., Cong WangZhejiang University, H., Peng ChengZhejiang University, H., \& Jiming ChenZhejiang University, H. (2024, May 6). Efficient vertical federated unlearning via fast retraining. ACM Transactions on Internet Technology.
>
> | Metrics | Accuracy (%) |                 |                    |                    |                    |                    |                 |                 |
> |:-------:|:------------:|:---------------:|:------------------:|:------------------:|:------------------:|:------------------:|:---------------:|:---------------:|
> | Metrics |   Baselines  |     Retrain     |         FT         |       Fisher       |      Amnesiac      |        Unsir       |        BU       |       Ours      |
> |    Dr   |     97.92    | 98.81 &pm; 0.34 |   81.89 &pm; 0.82  |   30.26 &pm; 0.21  |   73.29 &pm; 0.09  |   64.19 &pm; 0.22  | 45.30 &pm; 0.91 | **89.05 &pm; 0.61** |
> |    Du   |     95.67    |  0.00 &pm; 0.00 | 4.33 &pm; 0.49 | **0.00 &pm; 0.00** | **0.00 &pm; 0.00** | **0.00 &pm; 0.00** |  3.67 &pm; 0.14 |  2.43 &pm; 0.04 |
>
> Table 1 : Single-label unlearning scenario with Brain MRI datasets on ResNet18 architecture. This experiments have one active party and two passive parties. The image features is split to half and each passive party own half of the image features. We unlearn label 0 (glioma) in this experiments.
> $\newline$
>
> **2)The discussion and presentation of experiment results can be improved. e.g., the paper discusses utility in Sec 5.2.1 but also mentioning effectiveness in Figure 4 in Sec 5.2.2. It is not intuitive to compare different methods on these two aspects easily.**
>
> Thank you for your constructive suggestion. Section 5.2.1 presents the model utility on the remained data $D_r$ through model accuracy, while Section 5.2.2 evaluates the unlearning effectiveness using the model accuracy on the unlearned data $D_u$ and the Attack Success Rate (ASR) of membership inference attacks (as defined in the evaluation metric of the main text). In the future version of the paper, \textbf{we will consolidate these three evaluation metrics into a single graph or table for better comparison.} Experimental results in Sect. 5 of the main text show our methods performs well in both model utility and unlearning effectiveness.
>
> $\newline$
>
> **3)Figure 1 is not easy to understand and interpret the label leakage risk.**
>
> We acknowledge the lack of clarity in our previous explanation. Figure 1 illustrates the risk of label leakage during the unlearning process in Vertical Federated Learning (VFL). In this process, the active party aims to unlearn the label information stored in the passive model owned by the passive party. The active party has to transfer certain information, such as gradients, to the passive party. However, this exchange may inadvertently reveal sensitive information, allowing the passive party to infer the label of the active party.

---

> ### Author Response · Authors · 2024-11-22
>
> **4)VFU seems to be used without definition**
>
> We appreciate the opportunity to clarify the previous explanation.
> Label unlearning in vertical federated learning (VFL) refers to the process of efficiently and securely removing label information from a VFL system.
> Specifically, the unlearned passive model, denoted as $\theta^u $, and the unlearned active model, denoted as $\omega^u $, are obtained through the application of an unlearning mechanism $\mathcal{U} $, as follows:
> \begin{equation*}
>   \theta^u = \mathcal{U}(\theta, g_u), \quad \omega^u = \mathcal{U}(\omega, y_u),
> \end{equation*}
> where $\theta $ and $\omega $ represent the passive and active models before unlearning, respectively, and $g_u $ are the gradients associated with the unlearned label $y_u $.
>
> Building upon the principles of machine unlearning presented in [2], label unlearning in VFL needs to satisfy the following three objectives:
>
> 1) **Selective Removal**: The influence of specific labels must be erased while preserving the integrity of other data.
> 2) **Efficiency**: The unlearning process should achieve the above without requiring the computational cost of retraining the model from scratch.
> 3) **Privacy Preservation**: The unlearning process must ensure that no sensitive label information is leaked to the passive party.
>
> We will incorporate this explanation into Section 3.1 of the paper.
>
> [2] Bourtoule L, Chandrasekaran V, Choquette-Choo C A, et al. Machine unlearning[C]//2021 IEEE Symposium on Security and Privacy (SP). IEEE, 2021: 141-159.
>
>
>
> **5)Can the proposed method be applied to non-vision datasets or other vision tasks other than classification?**
>
> Thank you for your suggestions. We have added experiments on Non-vision dataset (Yahoo Answers dataset [3]) for the classification task. Yahoo Answers is a dataset designed for text classification tasks, comprising 10 classes (topics) such as "Society \& Culture," "Science \& Mathematics," "Health," "Education \& Reference," among others. Each class contains 140,000 training samples and 6,000 testing samples. For simplicity, we utilized 5,000 training samples and 2,000 testing samples from each class.
>
> Table 2 below illustrates that our method performs effectively on both the accuracy of the remained data and the unlearned data. For instance, the unlearned data accuracy decreases from 41.63\% to 5.14\%, while the accuracy drop on the remained data is less than 3\%.
> | Metrics | Accuracy (%) |                 |                    |
> |:-------:|:------------:|:---------------:|:------------------:|
> | Metrics |   Baselines  |     Retrain     |      Ours      |
> |    $\mathcal{D}_{r}$   |     62.92    | 63.14 &pm; 0.45 |  60.72 &pm; 0.98 |
> |     $\mathcal{D}_{u}$    |     41.63    |  0.00 &pm; 0.00 |  5.14 &pm; 1.04 |
>
> Table 2 : Single-label unlearning scenario on Yahoo Answer datasets with MixText architecture ([4], transformer-based models). This experiments have one active party and two passive parties. Each sample (a single paragraph of text) is divided into two paragraphs, with each passive party holding one of them. We unlearn label 6 (Business \& Finance) in this experiments.
>
> [3] Fu, C., Zhang, X., Ji, S., Chen, J., Wu, J., Guo, S., Zhou, J., Liu, A. X., \& Wang, T. (1970, January 1). Label inference attacks against Vertical Federated Learning. USENIX.
>
> [4]Jiaao Chen, Zichao Yang, and Diyi Yang. 2020. MixText: Linguistically-Informed Interpolation of Hidden Space for Semi-Supervised Text Classification. In Proceedings of the 58th Annual Meeting of the Association for Computational Linguistics, pages 2147–2157, Online. Association for Computational Linguistics.

---

> ### Author Response · Authors · 2024-12-02
> **Follow up rebuttal**
>
> As today marks the final opportunity for reviewers to share any additional feedback with the authors, I wanted to kindly follow up to check if you have any further questions or require additional clarifications regarding our rebuttal. We would be delighted to address any remaining concerns promptly within the timeframe available (by tomorrow).
>
> We hope our responses have effectively addressed your valuable feedback. If they meet your expectations, we would greatly appreciate it if you could reflect this in your rating.
>
> Once again, we deeply appreciate the time, effort, and expertise you have dedicated to reviewing our work. Your constructive comments have been instrumental in enhancing the quality of our submission, and we are truly grateful for your guidance throughout this process.
>
> Thank you for your continued support.

---

### Author Response · Authors · 2024-11-28
**Rebuttal Summary + Revised PDF**

**Dear All Reviewers:**

We sincerely appreciate all reviewers' thoughtful feedbacks. Accordingly, we have included detailed explanations and clarifications in the respective rebuttal sections for each reviewer. We hope our responses adequately address your concerns. Thank you again for your feedback.

Also, we have carefully revised our manuscript accordingly. The following modifications have been incorporated into the revised PDF:

1) **Experiments on Non-Vision and Healthcare Datasets:** We have conducted additional experiments using the Brain Tumor MRI and Yahoo Answers datasets. The results of these experiments are detailed in Appendix A.4.

2) **Discussion on Unlearning Effectiveness:** A mathematical analysis is now added in Appendix A.1 to further demonstrate the unlearning effectiveness of the proposed method.

3) **Baselines:** The implementation details of baselines is transfered from appendix to the main paper - Section 5.1 **Baselines** at page 7.

4) **Ablation Study:**
        a)  **More passive parties:** Additional experiments are conducted with more passive parties in a label unlearning scenario to showcase the scalability of our proposed method. Moreover, we compare the unlearning time of various unlearning methods with different numbers of passive parties.


        b) **Effectiveness on $\lambda$:** A new ablation study on the impact of the parameter $\lambda$ is included in Appendix A.4.




5) **Clarifications and Revisions:**
        a) **Figures:** Figure 1 is redrawn, and explanations are added to Figures 1 and 6 (now Figure 5 in revised pdf).

        b) **Definitions and Notations:** The definition of VFU is now included in Section 3.1, and a notation table is added to Appendix A.2.

        c) **Experimental Details:** Additional experimental details are provided in Appendix A.3.

        d) **ASR Metrics:** ASR graph figures are replaced with corresponding ASR scores in Tables 1-4 for improved clarity and comparison.

---

### Meta-Review · Area_Chair_kVnK · 2024-12-20

**Metareview:**

This paper received two positive and two negative reviews. Positive reviews highlighted its clear structure and effective mixup augmentation method for unlearning in Vertical Federated Learning (VFL). Empirical results show improved accuracy and reduced attack success rates, with the method being practical for few-shot settings and applicable across datasets and models. It integrates well with techniques like differential privacy and gradient compression, achieving the lowest unlearning cost compared to existing methods. However, the negative reviews raised several concerns. The lack of a section explaining mathematical symbols makes the methodology hard to follow, and definitions for variables are needed. While the use of gradient ascent and inverse gradients is justified, the paper does not explain why these methods are effective for label removal or discuss the convergence conditions. An analysis of the forgetting mechanism would strengthen the paper. The experimental setup lacks detail, such as categories for forgetting, client numbers, and data distribution, which hinders reproducibility. A table summarizing this information is recommended. Furthermore, while VFL is used in fields like finance and healthcare, the experiments focus on simple vision datasets, which may not capture real-world complexities. The presentation of results could be clearer, particularly in comparing utility and effectiveness. The authors have made considerable efforts in their response, the Area Chair (AC) believes that the paper requires significant revisions for ICLR publication, particularly in terms of experimental analysis.

**Additional Comments On Reviewer Discussion:**

Two negative reviewers did not participate in the review discussion, two positive reviewer have recognized the rebuttal and revision.

1. Motivation/problem definition issues. Motivation is privacy such as GDPR & CCPA, for which instance-wise unlearning is the relevant goal. However, the method and experiments are designed for class-wise unlearning. Class-wise unlearning is not the mainstream, and it’s not explained why it’s important.
2. The data used in the main experiments is not reflective of the typical finance or healthcare settings where VFL is used. A little bit of this is added in the rebuttal, but this is a too major late revision.
3. Lots of unclear details, including lack of clarity on how the baselines are adapted to VFL setting (besides missing details mentioned by others).
4. Methodological Novelty. In the end the solution is to use a fairly standard few-shot learning augmentation.

---

### Decision · Program_Chairs · 2025-01-22

Reject